# Sparse identification of nonlinear dynamics with Shallow Recurrent Decoder Networks

## Abstract

Spatio-temporal modeling of real-world data is a challenging problem as a result of inherent high-dimensionality, measurement noise, and expensive data collection procedures. In this paper, we present **S**parse **I**dentification of **N**onlinear **Dy**namics with **SH**allow **R**ecurrent **D**ecoder networks (SINDy-SHRED) to jointly solve the sensing and model identification problems with simple implementation, efficient computation, and robust performance. SINDy-SHRED utilizes Gated Recurrent Units (GRUs) to model the temporal sequence of sensor measurements along with a shallow decoder network to reconstruct the full spatio-temporal field from the latent state space using only a few available sensors. Our proposed algorithm introduces a SINDy-based regularization. Beginning with an arbitrary latent state space, the dynamics of the latent space progressively converges to a SINDy-class functional, provided the projection remains within the set. We conduct a systematic experimental study including synthetic PDE data, real-world sensor measurements for sea surface temperature, and direct video data. With no explicit encoder, SINDy-SHRED allows for efficient training with minimal hyperparameter tuning and laptop-level computing. SINDy-SHRED demonstrates robust generalization in a variety of applications with minimal to no hyperparameter adjustments. Additionally, the interpretable SINDy model of latent state dynamics enables accurate long-term video predictions, achieving state-of-the-art performance and outperforming all baseline methods considered, including Convolutional LSTM, PredRNN, ResNet, and SimVP.

## 1 Introduction

Modeling unknown physics is an exceptionally challenging task that is complicated further by the computational burden of high-dimensional state spaces and expensive data collection. Partial differential equations (PDEs) derived from first principles remain the most ubiquitous class of models to describe physical phenomena. However, we frequently find that the simplifying assumptions necessary to construct a PDE model can render it ineffectual for real data where the physics is multi-scale in nature, only partially known, or where first principles models currently do not exist. In such cases, machine learning (ML) method offers an attractive alternative for learning both the physics and coordinates necessary for quantifying observed spatio-temporal phenomenon. Many recent efforts utilizing ML techniques seek to relax the computational burden for PDE simulation by learning surrogate models to forward-simulate or predict spatiotemporal systems. However, this new machine learning paradigm frequently exhibits instabilities during the training process, unstable roll outs when modeling future state predictions, and often yields minimal computational speedups.

Shallow Recurrent Decoder networks (SHRED) (Williams et al., 2024) are a recently introduced architecture that utilize data from sparse sensors to reconstruct and predict the entire spatiotemporal domain. Similar to Takens' embedding theorem, SHRED models trade spatial information at a single time point for a trajectory of sensor measurements across time. Previous work has shown SHRED can achieve excellent performance in examples ranging from weather forecasting, atmospheric ozone concentration modeling, and turbulent flow reconstructions. In this paper, we introduce **S**parse **I**dentification of **N**onlinear **Dy**namics with **SH**allow **R**ecurrent **D**ecoder networks (SINDy-SHRED). SINDy-SHRED exploits the latent space of recurrent neural networks for sparse sensor modeling, and enforces interprebility via a SINDy-based functional class. In this way, SINDy-SHRED enables a robust and sample-efficient joint discovery of the governing equation and

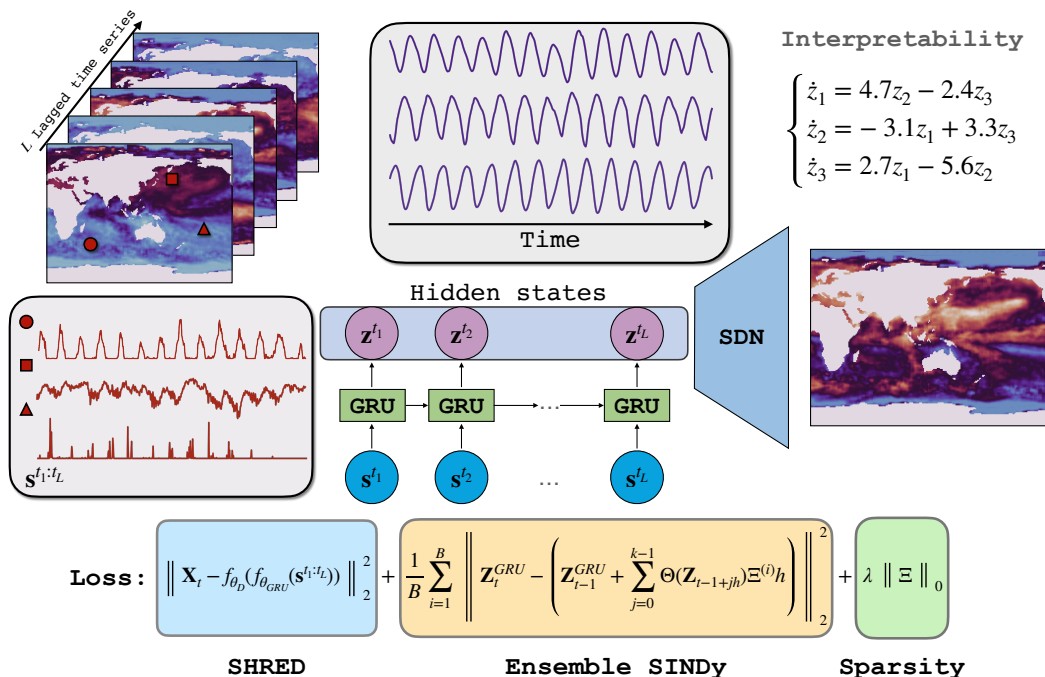

Figure 1: Illustration of the SINDy-SHRED architecture. SINDy-SHRED transfers the original sparse sensor signal (red) to an interpretable latent representation (purple) that falls into the SINDy-class functional. The shallow decoder performs a reconstruction in the pixel space.

coordinate system. With the correct governing equation, SINDy-SHRED can perform an accurate long-term prediction in a learned latent space, and in turn allow for long-term forecasting in the pixel space. For practical applications, SINDy-SHRED is a lightweight model which can perform low-rank recovery with only a few (e.g. three) active sensors, which is critical for large-scale scientific data modeling and real-time control. It does not require large amounts of data during training, thereby avoiding a common pitfall in existing ML techniques for accelerating physics simulations. SINDy-SHRED also exhibits remarkable training speed, even when executed on a single laptop. Furthermore, SINDy-SHRED is highly reproducible with minimal effort in hyperparameter tuning. The recommended network structure, hyperparameter, and training setting can generalize to many different datasets. In short, we demonstrate SINDy-SHRED to be very robust and highly applicable in many modern scientific modeling problems.

Existing work seeking to perform data-driven, long-term forecasts of spatio-temporal phenomena typically suffers from (i) instabilities and (ii) massive computational requirements. We find that SINDy-SHRED ameliorates many of these issues because (i) it is based on a stable equation discovery and (ii) the learned model is an ODE in a learned latent space, rendering simulation more computationally efficient. We further conjecture that the fact SINDy-SHRED does not include a spatial encoder contributes to the architecture's robustness, rendering it more difficult for the model to overfit during training.

We perform a wide range of studies to demonstrate the effectiveness of SINDy-SHRED. We first apply the model on the sea surface temperature data, which is a complex real-world problem. We also consider data from a complex simulation of atmospheric chemistry, video data of flow over a cylinder, and video data of a pendulum. The ability of SINDy-SHRED to perform well on video data is an important result for so-called "GoPro physics." With extremely small sample size and noisy environments, SINDy-SHRED achieves governing equation identification with stable long-term predictions. Finally, we demonstrate the performance of SINDy-SHRED on a chaotic 2D Kolmogorov flow (in Appendix D), finding a reasonable model even in the presenece of chaos. The contribution of our paper is three-fold.

- We propose SINDy-SHRED to incorporate symbolic understanding of the latent space of recurrent models of spatio-temporal dynamics.
- We further analyze the latent space of case studies and propose scientific models for these systems.
- We systematically study SINDy-SHRED and compare to popular deep learning algorithms in spatio-temporal prediction.

## 2 RELATED WORKS

Traditionally, spatio-temporal physical phenomena are modeled by Partial Differential Equations (PDEs). To accelerate PDE simulations, recent efforts have leveraged neural networks to model physics. By explicitly assuming the underlying PDE, physics-informed neural networks (Raissi et al., 2019) utilize the PDE structure as a constraint for small sample learning. However, assuming the exact form of governing PDE for real data can be a strong limitation. There have been many recent works on learning and predicting PDEs directly using neural networks (Khoo et al., 2021; Li et al., 2020b; Holl et al., 2020; Lu et al., 2021; Lin et al., 2021). Meanwhile, PDE-find (Rudy et al., 2017; Messenger & Bortz, 2021; Fasel et al., 2022) offers a data-driven approach to identify PDEs from the spatial-temporal domain. Still, the high-dimensionality and required high data quality can be prohibitive for practical applications.

In parallel, previous efforts in the discovery of physical law through dimensionality reduction techniques (Champion et al., 2019; Lusch et al., 2018; Mars Gao & Nathan Kutz, 2024) provide yet another perspective on the modeling of scientific data. The discovery of physics from a learned latent space has previously been explored by (Fukami et al., 2021; Cheng et al., 2024; Farenga et al., 2024; Conti et al., 2023; Wu et al., 2022; Li et al., 2020a), yet none of these methods consider a regularization on the latent space with no explicit encoder. Yu et al. proposed the idea of physics-guided learning (Yu & Wang, 2024) which combines physics simulations and neural network approximations. Directly modeling physics from video is also the subject of much research in the field of robotics (Finn et al., 2016; Todorov et al., 2012; Sanchez-Gonzalez et al., 2018), computer vision (Xie et al., 2024; Wu et al., 2017) and computer graphics (Kandukuri et al., 2020; Liu et al.; Wu et al., 2015; Mrowca et al., 2018), since many fields of research require better physics models for simulation and control. From the deep learning side, combining the structure of differential equations into neural networks (He et al., 2016; Chen et al., 2018) has been remarkably successful in a wide range of tasks. When spatial-temporal modeling is framed as a video prediction problem, He et al. found (He et al., 2022) that random masking can be an efficient spatio-temporal learner, and deep neural networks can provide very good predictions for the next 10 to 20 frames (Shi et al., 2015; Wang et al., 2017; Gao et al., 2022; Guen & Thome, 2020). Generative models have also been found to be useful for scientific data modeling (Mirza, 2014; Song et al., 2021; Cachay et al., 2024).

## 3 METHODS

The shallow recurrent decoder network (**SHRED**) is a computational technique that utilizes recurrent neural networks to predict the spatial domain. (Williams et al., 2024). The method functions by trading high-fidelity spatial information for trajectories of sparse sensor measurements at given spatial locations. Mathematically, consider a high-dimensional data series $\{\boldsymbol{X}_i\}_{i=1}^T \in \mathbb{R}^{(W \times H) \otimes T}$ that represents the evolution of a spatio-temporal dynamical system, $W$, $H$, and $T$ denote the width, height, and total time steps of the system, respectively. In SHRED, each sensor collects data from a fixed spatial position in a discretized time domain. Denote the subset of sensors as $\mathcal{S}$, the input data of SHRED is $\{\boldsymbol{X}^{\mathcal{S}}\}_{i=1}^T \in \mathbb{R}^{\text{card}(S) \otimes T}$. Provided the underlying PDE allows spatial information to propagate, these spatial effects will appear in the time history of the sensor measurements, enabling the sensing of the entire field using only a few sensors. In vanilla SHRED, a Long Short-Term Memory (LSTM) module is used to map the sparse sensor trajectory data into a latent space, followed by a shallow decoder to reconstruct the entire spatio-temporal domain at the current time step.

SHRED enables efficient sparse sensing that is widely applicable to many scientific problems (Ebers et al., 2024; Kutz et al., 2024; Riva et al., 2024). The advantage of SHRED comes from three aspects. First, SHRED only requires minimal sensor measurements. Under practical constraints, collecting full-state measurements for data prediction and control can be prohibitively expensive.

Second, SHRED does not require grid-like data collection, which allows for generalization to more complex data structures. For example, it is easy to apply SHRED to graph data with an unknown underlying structure, such as human motion data on joints, robotic sensor data, and financial market data. Moreover, SHRED is theoretically rooted in PDE modeling methods from the perspective of separation of variables, which has the potential to offer strong theoretical guarantees such as convergence and stability.

### 3.1 EMPOWERING SHRED WITH REPRESENTATION LEARNING AND PHYSICS DISCOVERY

To achieve a parsimonious representation of physics, it is important to find a representation that effectively captures the underlying dynamics and structure of the system. In SINDy-SHRED (shown in Fig. 1), we extend the advantages of SHRED, and perform a joint discovery of coordinate systems and governing equations. This is accomplished by enforcing that the latent state of the recurrent network follows an ODE in the SINDy class of functions.

**Finding better representations**   SHRED has a natural advantage in modeling the latent governing physics due to its small model size. SHRED is based on a shallow decoder with a relatively small recurrent network structure. The relative simplicity of the model allows the latent representation to maintain many advantageous properties such as smoothness and Lipschitzness. Experimentally, we observe that the hidden state space of a SHRED model is generally very smooth. Second, SHRED does not have an explicit encoder, which avoids the potential problem of spectral bias (Rahaman et al., 2019). Many reduced-order modeling methods that rely on an encoder architecture struggle to learn physics and instead focus only on modeling the low-frequency information (background) (Refinetti & Goldt, 2022; Champion et al., 2019; Mars Gao & Nathan Kutz, 2024). Building upon SHRED, we further incorporate SINDy to regularize the learned recurrence with a well-characterized and simple form of governing equation. In other words, we perform a joint discovery of a coordinate system (which transfers the high-dimensional observation into a low-dimensional representation) and the governing law (which describes how the summarized latent representation progresses forward with respect to time) of the latent space of a SHRED model. This approach is inspired by the principle in physics that, under an ideal coordinate system, physical phenomena can be described by a parsimonious dynamical model (Champion et al., 2019; Mars Gao & Nathan Kutz, 2024). When the latent representation and the governing law are well-aligned, this configuration is likely to capture the true underlying physics. This joint discovery results in a latent space that is both interpretable and physically meaningful, enabling robust and stable future prediction based on the learned dynamics.

### 3.2 SINDY-SHRED: LATENT SPACE REGULARIZATION VIA SINDY

As a compressive sensing procedure, there exist infinitely many equally valid solutions for the latent representation. Therefore, it is not necessary for the latent representation induced by SHRED to follow a well-structured differential equation. For instance, even if the exhibited dynamics are fundamentally linear, the latent representation may exhibit completely unexplainable dynamics, making the model challenging to interpret and extrapolate. Therefore, in SINDy-SHRED, our goal is to further constrain the latent representations to lie within the SINDy-class functional. This regularization promotes models that are fundamentally explainable by a SINDy-based ODE, allowing us to identify a parsimonious governing equation. The SINDy class of functions typically consists of a library of commonly used functions, which includes polynomials, and Fourier series. Although they may seem simple, these functions possess surprisingly strong expressive power, enabling the model to capture very complex dynamical systems.

**SINDy as a Recurrent Neural network**   We first reformulate SINDy using a neural network form, simplifying its incorporation into a SHRED model. ResNet (He et al., 2016) and Neural ODE (Chen et al., 2018) utilize skip connections to model residual and temporal derivatives. Similarly, this could also be done via a Recurrent Neural Network (RNN) which has a general form of

$$z_{t+1} = z_t + f(x_t),\qquad(1)$$

where $f(\cdot)$ is some function of the input. From the Euler method, the ODE forward simulation via SINDy effectively falls into the category of Recurrent Neural Networks (RNNs) which has the form

$$z_{t+1} = z_t + f_\Theta(x_t, \Xi, \Delta t),\qquad(2)$$

where $f_\Theta(x_t, \Xi, \Delta t) = \Theta(x_t)\Xi\Delta t$ is a nonlinear function. Notice that this $f_\Theta(\cdot)$ has exactly the same formulation as in SINDy (Brunton et al., 2016). The application of function libraries with sparsity constraints is a manner of automatic neural architecture search (NAS) (Zoph & Le, 2016). Compared to all prior works (Champion et al., 2019; Fukami et al., 2021; Conti et al., 2023), this implementation of the SINDy unit fits better in the framework of neural network training and gradient descent. We utilize trajectory data $\{\mathbf{z}_i\}_{i=1}^T$ and forward simulate the SINDy-based ODE using a trainable parameter $\Xi$. To achieve better stability and accuracy for forward integration, we use Euler integration with $k$ mini-steps (with time step $\frac{\Delta t}{k}$) to obtain $\mathbf{z}_{t+1}$. In summary, defining $h = \frac{\Delta t}{k}$, we optimize $\Xi$ with the following:

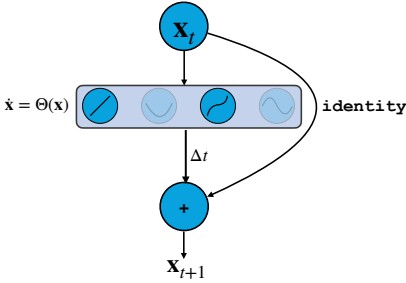

Figure 2: Diagram of the RNN form of SINDy.

$$\Xi = \arg\min \left\| \mathbf{z}_{t+1} - \left( \mathbf{z}_t + \sum_{i=0}^{k-1} \Theta(\mathbf{z}_{t+ih})\Xi h \right) \right\|_2^2, \quad \mathbf{z}_{t+ih} = \mathbf{z}_t + \Theta(\mathbf{z}_{t+(i-1)h})\Xi h, \quad \min \|\Xi\|_0.$$

(3)

To achieve $\ell_0$ optimization, we perform pruning with $\ell_2$ which approximates $\ell_0$ regularization under regularity conditions (Zheng et al., 2014; Gao et al., 2023; Blalock et al., 2020). Applying SINDy unit has the following benefits: (a) The SINDy-function library contains frequently used functions in physics modeling (e.g. polynomials and Fourier series). (b) With sparse system identification, the neural network is more likely to identify governing physics, which is fundamentally important for extrapolation and long-term stability.

**Latent space regularization via ensemble SINDy** We first note that we deviate from the original SHRED architecture by using a GRU as opposed to an LSTM. This choice was made because we generally found that GRU provides a smoother latent space. Now, recall that our goal is to find a SHRED model with a latent state that is within the SINDy-class functional. However, the initial latent representation from SHRED does not follow the SINDy-based ODE structure at all. On the one hand, if we naively apply SINDy to the initial latent representation, the discovery is unlikely to fit the latent representation trajectory. On the other hand, if we directly replace the GRU unit to SINDy and force the latent space to follow the discovered SINDy model, it might lose information that is important to reconstruction the entire spatial domain. Therefore, it is important to let the two latent spaces align progressively.

In Algorithm 1, we describe our training procedure that allows the two trajectories to progressively align with each other. To further ensure a gradual adaptation and avoid over-regularization, we introduce ensemble SINDy units with varying levels of sparsity constraints, which ranges the effect from promoting a full model (all terms in the library are active) to a null model (where no dynamics are represented). From the initial latent representation $\mathbf{z}_{1:t}^{\text{iter } 0}$ from SHRED, the SINDy model first provides an initial estimate of ensemble SINDy coefficients $\{\hat{\Xi}_0^i\}_{i=b}^B$. Then, the parameters of SHRED will be updated towards the dynamics simulated by $\{\hat{\Xi}_0\}_{i=b}^B$, which generates a new latent representation trajectory $\mathbf{z}_{1:t}^{\text{iter } 1}$. We iterate this procedure and jointly optimize the following loss function to let the SHRED latent representation trajectory approximate the SINDy generated trajectory:

$$\mathcal{L} = \left\| \mathbf{X}_t - f_{\theta_D}(f_{\theta_{\text{GRU}}}(\mathbf{X}_{t-L:t}^{\mathcal{S}})) \right\|_2^2 + \sum_{i=1}^B \left\| \mathbf{Z}_t^{\text{GRU}} - \left( \mathbf{Z}_{t-1}^{\text{GRU}} + \sum_{j=0}^{k-1} \Theta(\mathbf{Z}_{t-1+jh})\Xi^{(i)}h \right) \right\|_2^2 + \lambda \|\Xi\|_0,$$

(4)

where $\mathbf{Z}_{t-1+ih} = \mathbf{Z}_t + \Theta(\mathbf{Z}_{t-1+(i-1)h})\Xi h$, $\mathbf{Z}_{t-1} = \mathbf{Z}_{t-1}^{\text{GRU}}$, and $h = \frac{\Delta t}{k}$.

## 4 EXPERIMENT

In the following, we perform case studies across a range of scientific and engineering problems.

---

**Algorithm 1** Latent state space regularization via SINDy

---

**Input:** input $\boldsymbol{X}^{\mathcal{S}}_{t-L:t+1}$, $\boldsymbol{X}_t$, SINDy library $\Theta(\cdot)$, timestep $\Delta t$.

1: **function** LATENTSPACESINDY($\boldsymbol{X}^{\mathcal{S}}_{t-L:t+1}$, $\boldsymbol{X}_{t+1}$, $\Delta t$)
2:     **for** i in $0, 1, \cdots, n-1$: **do**
3:         $\boldsymbol{Z}_t$, $\boldsymbol{Z}_{t+1} = f_{\theta_{\text{GRU}}}(\boldsymbol{X}^{\mathcal{S}}_{t-L:t})$, $f_{\theta_{\text{GRU}}}(\boldsymbol{X}^{\mathcal{S}}_{t-L+1:t+1})$;
4:         **for** j in $(0, 1, \frac{\Delta t}{k})$: **do**             ▷ SINDy forward simulation
5:             $\boldsymbol{Z}^{\text{SINDy}}_{t+\frac{j+1}{k}\Delta t} = \boldsymbol{Z}^{\text{SINDy}}_{t+\frac{j}{k}\Delta t} + \Theta(\boldsymbol{Z}^{\text{SINDy}}_{t+\frac{j}{k}\Delta t})\Xi\Delta t$
6:         **end for**
7:         $\hat{\boldsymbol{X}}_{t+1} = f_{\theta_D}(\boldsymbol{Z}_{t+1})$             ▷ SHRED reconstruction
8:         $\theta_{\text{GRU}}, \Xi, \theta_D = \arg\min_{\theta_{\text{GRU}},\Xi,\theta_D} \left\| \mathbf{X}_{t+1} - \hat{\boldsymbol{X}}_{t+1} \right\|_2^2 + \left\| \mathbf{Z}^{\text{GRU}}_{t+1} - \boldsymbol{Z}^{\text{SINDy}}_{t+1} \right\|_2^2 + \lambda \left\| \Xi \right\|_0$
9:         **if** $i \mod 100 = 0$ **then**
10:             $\Xi[|\Xi| < \text{threshold}] = 0$
11:         **end if**
12:     **end for**                              ▷ Train until converges
13: **end function**

---

**Sea-surface temperature** The first example we consider is that of global sea-surface temperature. The SST data contains 1,400 weekly snapshots of the weekly mean sea surface temperature from 1992 to 2019 reported by NOAA (Reynolds et al., 2002). The data is represented by a $180 \times 360$ grid, of which 44,219 grid points correspond to sea-surface locations. We standardize the data with its own min and max, which transforms the sensor measurements to within the numerical range of $(0, 1)$. We randomly select 250 sensors from the possible 44,219 locations and set the lag parameter to 52 weeks. The inclusion of 250 sensors is a substantial deviation from previous work with SHRED in which far fewer sensors were used Williams et al. (2024). However, we found greater robustness in the application of E-SINDy to the learned latent state when more sensors were utilized. Thus, for each input-output pair, the input consists of the 52-week trajectories of the selected sensors, while the output is a single temperature field across all 44,219 spatial locations. SINDy-SHRED aims to reconstruct the entire sea surface temperature locations from these randomly selected sparse sensor trajectories. We include the details of the experimental settings of SINDy-SHRED in the Appendix C.1. From the discovered coordinate system, we define the representation of physics - the latent hidden state space - to be $(z_1, z_2, z_3)$. The dynamics progresses forward via the following set of equations:

$$\begin{cases} \dot{z}_1 &= 4.68z_2 - 2.37z_3, \\ \dot{z}_2 &= -3.10z_1 + 3.25z_3, \\ \dot{z}_3 &= 2.72z_1 - 5.55z_2. \end{cases} \quad (5)$$

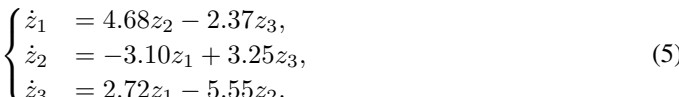

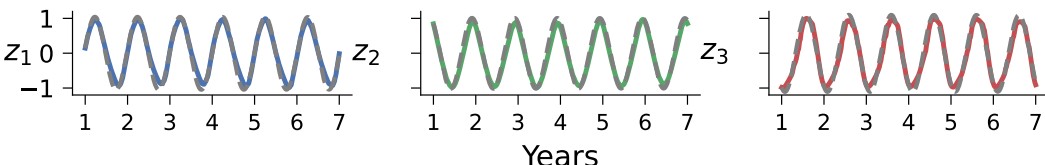

Figure 3: Extrapolation of latent representation in SINDy-SHRED from the discovered dynamical system for SST. Colored: true latent representation. Grey: SINDy extrapolation.

The discovery of a linear system describing the evolution of the latent state is in line with prior work on SST data De Bézenac et al. (2019) in which it as assumed that the underlying physics is an advection-diffusion PDE. In Fig. 3 (a) we further present the accuracy of this discovered system by forward simulating the system from an initial condition for a total of 27 years (c.f. Fig. 15). It is observable how the discovered law is close to the true evolution of latent hidden states and, critically, there appears to be minimal phase slipping. Extrapolating the latent state space via forward integration, we can apply the shallow decoder to return forecasts of the high-dimensional data. Doing

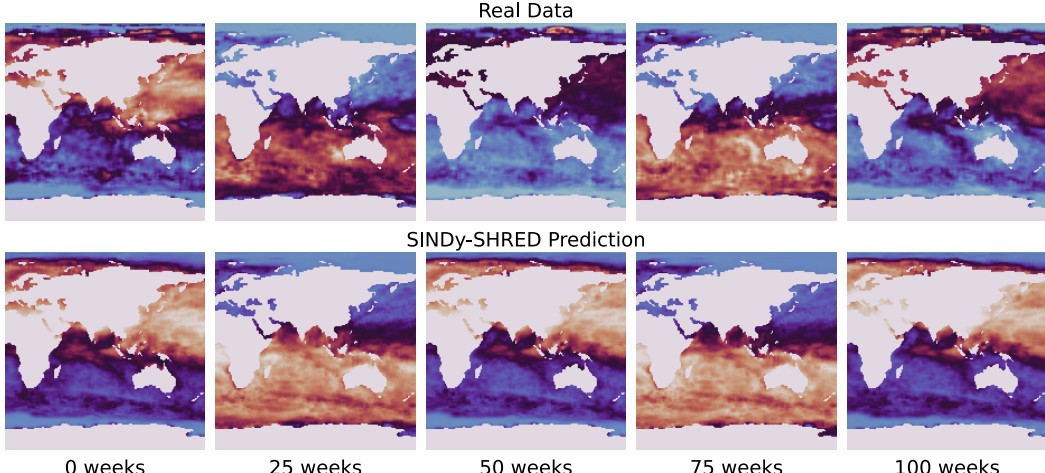

Figure 4: Long-term global sea-surface temperature prediction via SINDy-SHRED from week 0 to week 100. We crop the global temperature map for better visualization.

so, we find an averaged MSE error of $0.57 \pm 0.10^{\circ}C$ for all prediction lengths in the test dataset. In Fig. 4, we show SINDy-SHRED produces stable long-term predictions for SST data. We further include Fig. 16 to demonstrate the extrapolation of each sensor. The sensor level prediction is based on the global prediction of the future frame, and we visualize the signal trajectory of specific sensor locations. We find SINDy-SHRED is robust for out-of-distribution sensors, though its extrapolation may not accurately capture anomalous events.

**3D Atmospheric ozone concentration** The atmospheric ozone concentration dataset (Bey et al., 2001) contains a one-year simulation of the evolution of an ensemble of interacting chemical species through a transport operator using GEOS-Chem. The simulation contains 1,456 temporal samples with a timestep of 6 hours over one year for 99,360 (46 by 72 by 30) spatial locations (latitude, longitude, elevation). The data presented in this work has compressed by performing an SVD and retaining only the first 50 POD modes. As with the SST data, we standardize the data within the range of $(0, 1)$ and randomly select and fix 3 sensors out of 99,360 spatial locations ($0.5\%$). We include the details of the experimental settings of SINDy-SHRED in the Appendix C.2. The converged latent representation presents the following SINDy model:

$$\begin{cases} \dot{z}_1 & = -0.002 - 0.013z_2 + 0.007z_3, \\ \dot{z}_2 & = -0.001z_1 + 0.004z_2 - 0.008z_3, \\ \dot{z}_3 & = 0.002 + 0.012z_2 - 0.005z_3. \end{cases} \quad (6)$$

The identified governing physics is close to a linear system with constant terms for damping. Unlike traditional architectures for similar problems, which may include expensive 3D convolution, SINDy-SHRED provides an efficient way of training, taking about half an hour. Although the quantity of data is insufficient to perform long term-predictions, SINDy-SHRED still exhibits interesting behavior for a longer-term extrapolation which converges to the fixed point at $\mathbf{0}$ (as shown in Fig. 17). From the extrapolation of the latent state space, the shallow decoder prediction has an averaged MSE error of $1.5e^{-2}$. In Fig. 6, we visualize the shallow decoder prediction up to 14 weeks. In Fig. 18, we reconstruct the sensor-level predictions which demonstrate the details of the signal prediction. The observations are much noiser than the SST data, but SINDy-SHRED provides a smoothed extrapolation for the governing trends.

**GoPro physics data: flow over a cylinder** In this subsection, we demonstrate the performance of SINDy-SHRED on an example of so-called "GoPro physics modeling." The considered data is collected from a dyed water channel to visualize a flow over a cylinder (Albright, 2017). The Reynolds number is 171 in the experiment. The dataset contains 11 seconds of video taken at 30 frames per second (FPS). We manually perform data augmentation and repeat the latter part of the video once to increase the number of available training samples . We transfer the original RGB

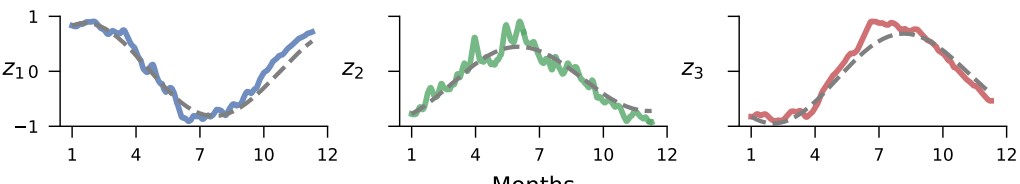

Figure 5: Extrapolation of latent representation in SINDy-SHRED from the discovered dynamical system for Ozone data. Colored: true latent representation. Grey: SINDy extrapolation.

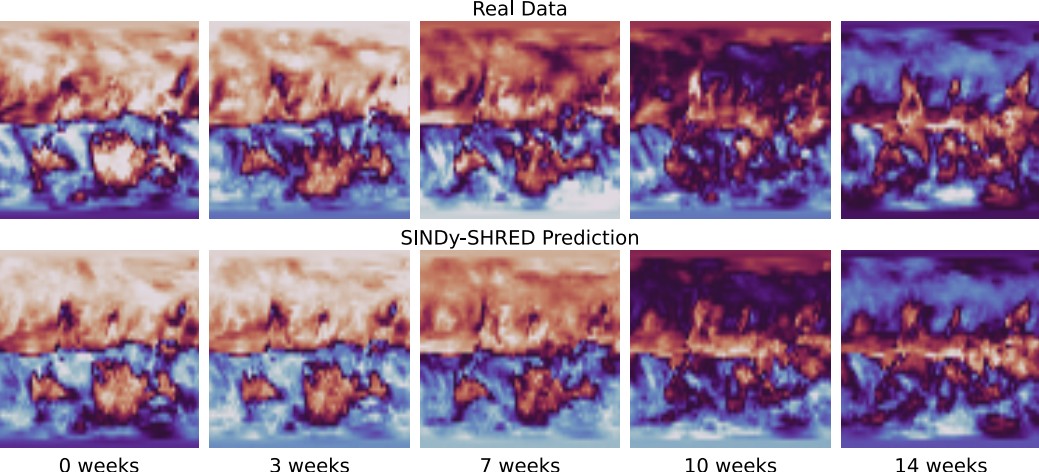

Figure 6: Long-term global Ozone data prediction via SINDy-SHRED with elevation 0 from week 0 to week 14.

channel to gray scale and remove the background by subtracting the mean of all frames. After the prior processing step, the video data has only one channel (gray) within the range $(0,1)$ with a height of 400 pixels and a width of 1,000 pixels. We randomly select and fix 200 pixels as sensor measurements from the entire $400,000$ space, which is equivalent to only $0.05\%$ of the data. We set the lag parameter to 60 frames. We include the details of the experimental settings of SINDy-SHRED in the Appendix C.4. We define the representation of the hidden latent state space as $(z_1, z_2, z_3, z_4)$. We discover the following dynamical system:

$$\begin{cases} \dot{z}_1 & = -0.69z_2 + 0.98z_3 - 0.40z_4, \\ \dot{z}_2 & = 1.00z_1 - 0.78z_31 - 0.31z_2z_3^2, \\ \dot{z}_3 & = -1.029z_1 + 0.59z_2 + 0.41z_4. \\ \dot{z}_4 & = -0.26z_1^2 - 0.29z_2^2z_3 - 0.39z_3^3. \end{cases} \quad (7)$$

Compared to the systems discovered in all previous examples, the flow over a cylinder model is much more complex with significant nonlinear interactions. In Eqn. 7, we find that $z_1$ and $z_3$ behave like a governing mode of the turbulence swing; $z_2$ and $z_4$ further depict more detailed nonlinear effects. We also present the result of extrapolating this learned representation. We generate the trajectory from the initial condition at time point 0 and perform forward integration for extrapolation. As shown in Fig. 7, the learned ODE closely follows the dynamics of $z_1$ and $z_3$ up to 7 seconds (210 timesteps); $z_2$ and $z_4$ also have close extrapolation up to 3 seconds.

This learned representation also nicely predicts the future frames in pixel space. The shallow decoder prediction has an averaged MSE error of $0.030$ (equivalently $3\%$) over the entire available trajectory. In Fig. 8, we observe that the autoregressively generated prediction frames closely follow the true data, and further in Fig. 21, we find that the predictions are still stable after 1,000 frames, which is out of the size of the original dataset. The sensor-level prediction in Fig. 20 further demonstrates the accuracy of reconstruction in detail.

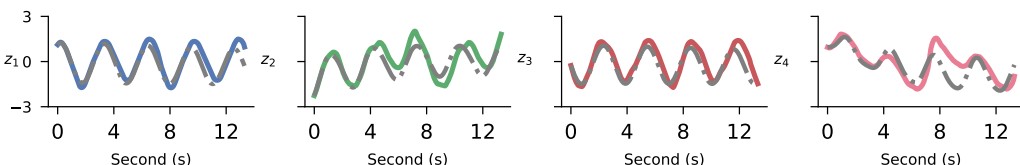

Figure 7: Extrapolation of latent representation in SINDy-SHRED from the discovered dynamical system for flow over a cylinder data. Colored: true latent representation. Grey: SINDy extrapolation.

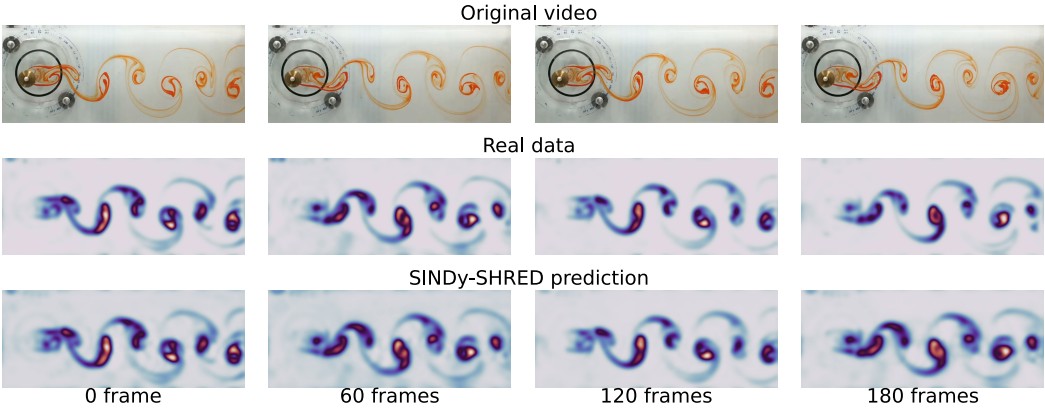

Figure 8: Long-term pixel space video prediction via SINDy-SHRED. We demonstrate the forward prediction outcome up to 180 frames.

**Baseline study: prediction of single shot real pendulum recording**    In this subsection, we compare the performance of SINDy-SHRED to other popular existing learning algorithms. We perform the baseline study particularly on video data of a pendulum since many deep learning algorithms are hard to scale up to deal with large scientific data. In the following, we demonstrate the result of video prediction on the pendulum data using ResNet (He et al., 2016), convolutional LSTM (convLSTM) (Shi et al., 2015), and PredRNN (Wang et al., 2017), and SimVP (Gao et al., 2022). The pendulum in our experiment is not ideal and includes complex damping effects. We use a nail on the wall and place the rod (with a hole) on the nail. This creates complex friction, which slows the rod more when passing the lowest point due to the increased pressure caused by gravity. The full model we discovered from the video (as shown in Fig. 9) includes four terms:

$$\ddot{z} = 0.17\dot{z}^2 - 0.06\dot{z}^3 - 10.87\sin(z) + 0.48\sin(\dot{z}). \tag{8}$$

As shown in Table 1, SINDy-SHRED outperforms all baseline methods for total error and long-term predictions. Generally, all baseline deep learning methods perform well for short-term forecasting, but the error quickly accumulatesfor longer-term predictions. This is also observable from the prediction in the pixel space as shown in Fig. 10. SINDy-SHRED is the only method that does not produce collapsed longer-term predictions. In Fig. 22, the sensor level prediction also demonstrates the robustness of the SINDy-SHRED prediction. PredRNN is the second best method as measured by the total error. However, PredRNN is expensive in computation which includes a complex forward pass with an increased number of parameters. It is also notable that the prediction of PredRNN collapses after 120 frames, after which only an averaged frame over the entire trajectory is predicted. ConvLSTM has a relatively better result in terms of generation, but the long-term prediction is still inferior compared to SINDy-SHRED. Additionally, we note that 2D convolution is much more computationally expensive. For larger spatiotemporal domains (e.g. the SST example and 3D ozone data), the computational complexity of convolution will scale up very quickly, which makes the algorithm impractical to execute. Similar computational issues will occur for diffusion models and generative models, which is likely to be impractical to compute, and unstable for longer-term predictions. In summary, we observe that SINDy-SHRED is not only a more accurate long-term model, but is also faster to execute and smaller in size.

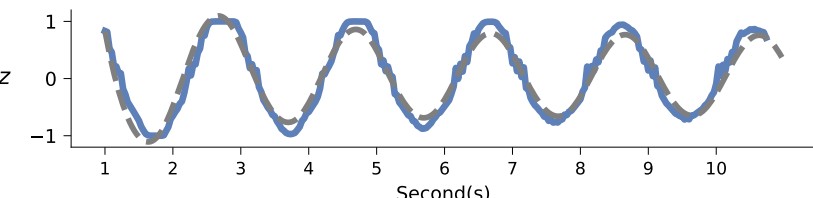

Figure 9: Extrapolation of latent representation in SINDy-SHRED from the discovered dynamical system for the pendulum moving data. Blue: true latent representation. Grey: SINDy extrapolation.

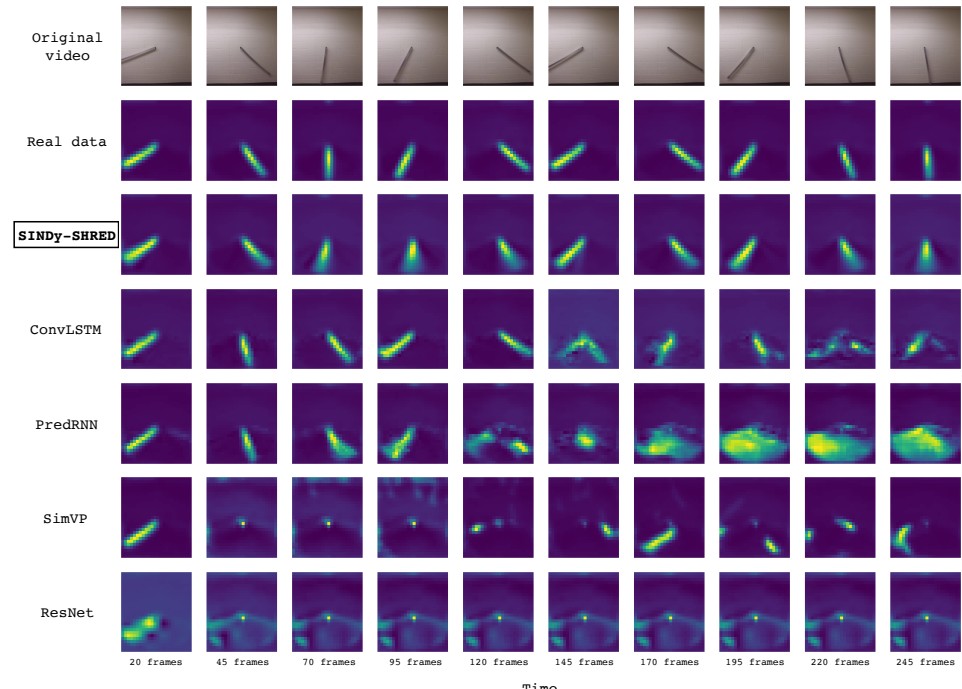

Figure 10: The pendulum video generation outcome from ResNet, SimVP, ConvLSTM, PredRNN, and SINDy-SHRED from frame 20 to frame 245.

| Models | Params # | Training time | $T = [0, 100]$ | $T = [100, 200]$ | $T = [200, 275]$ | Total |
|--------|----------|---------------|----------------|------------------|------------------|-------|
| ResNet (He et al., 2016) | 2.7M | 24 mins | $2.08 \times 10^{-2}$ | $1.88 \times 10^{-2}$ | $2.05 \times 10^{-2}$ | $2.00 \times 10^{-2}$ |
| SimVP (Gao et al., 2022) | 460K | 30 mins | $2.29 \times 10^{-2}$ | $2.47 \times 10^{-2}$ | $2.83 \times 10^{-2}$ | $2.53 \times 10^{-2}$ |
| PredRNN (Wang et al., 2017) | 444K | 178 mins | $1.02 \times 10^{-2}$ | $1.79 \times 10^{-2}$ | $1.69 \times 10^{-2}$ | $1.48 \times 10^{-2}$ |
| ConvLSTM (Shi et al., 2015) | 260K | 100 mins | $\mathbf{9.24 \times 10^{-3}}$ | $1.86 \times 10^{-2}$ | $1.99 \times 10^{-2}$ | $1.55 \times 10^{-2}$ |
| **SINDy-SHRED**[*] | **44K** | **17 mins** | $1.70 \times 10^{-2}$ | $\mathbf{9.36 \times 10^{-3}}$ | $\mathbf{5.31 \times 10^{-3}}$ | $\mathbf{1.05 \times 10^{-2}}$ |

Table 1: Comparison table of SINDy-SHRED to baseline methods for parameter size, training time, and mean-squared error over different prediction horizon.

## 5 CONCLUSION

In this paper, we present SINDy-SHRED, which jointly performs the discovery of coordinate systems and governing equations with low computational cost and strong predictive power. Through experiments, we show that our method can produce robust and accurate long-term predictions for a variety of complex problems, including global sea-surface temperature, 3D atmospheric ozone concentration, flow over a cylinder, and a moving pendulum. SINDy-SHRED achieves state-of-the-art performance in long-term autoregressive video prediction, outperforming ConvLSTM, PredRNN, ResNet, and SimVP with the lowest computational cost and training time.

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
