## A CHALLENGES IN ROLLING OUT NEURAL NETWORKS FOR FITTING A SIMPLE SINE FUNCTION

In the following example, we consider a simple use case in which we fit a simple sine function using recurrent neural networks. Surprisingly, extrapolating a simple sine function can be highly nontrivial for neural networks.

We implement a GRU network in the following. The GRU network consists of an input layer, three stacked GRU layers with size 500, and a fully connected output layer. We employ the Adam optimizer with a learning rate of 0.001 and used the mean squared error (MSE) as the loss function. We train the GRU network with 150 epochs with a batch size of 1. The input sequences are made up of 50 time steps, normalized to the range [0, 1].

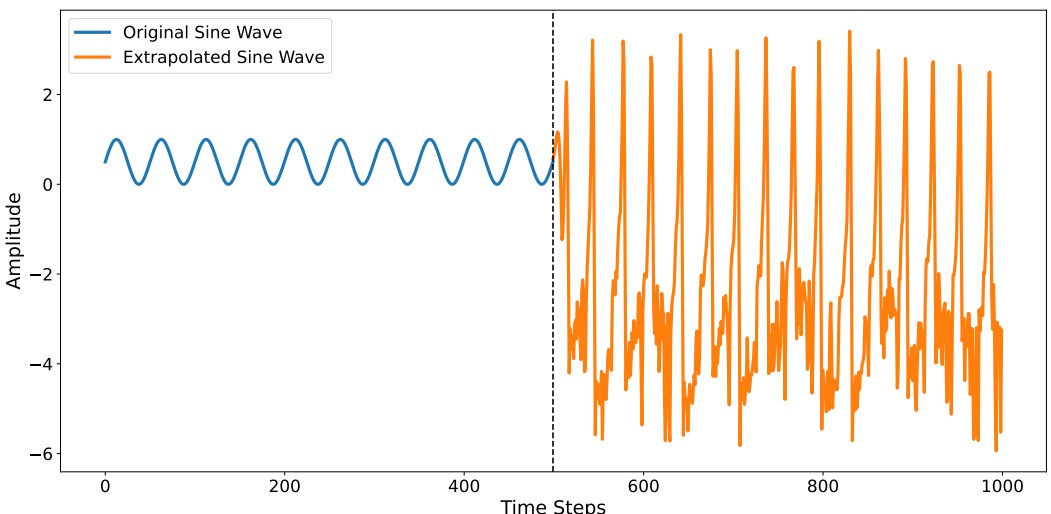

Figure 11: Fitting the sine function using a GRU network could be challenging.

Even though the training loss is near-optimal with $1e^{-7}$. However, the extrapolation is very poor. We note here that by carefully specifying the GRU network and making it a mostly linear unit (within the SINDy-class functional), the extrapolation result could be reasonable. But, the general setting of GRU networks, as shown in Fig. 11, fails to predict well. This motivating example demonstrates the fact that knowledge with physical meanings, e.g. the sine function here, is hard for neural networks to approximate accurately in extrapolation. Therefore, this demonstrates the necessity of integrating well-characterized ODE structures with non-linear functions into deep learning models, as they are important for accurate learning and prediction of physics.

## B QUALITATIVE RESULT ON THE ERROR BOUNDS

We further establish the statistical foundation for dynamical system learning. When the underlying dynamical system can be closely described by a linear combination of the library of functions, obtaining a "governing equation" will have huge benefits for long-term extrapolation. Due to the nature of forward integration,error accumulates rapidly making an approximate system undesirable for extrapolation. In the following, we formalize this statement by analyzing the Rademacher complexity of SINDy-class and neural networks functional.

The system we wish to study has the form that

$$\dot{x} = f(x), \tag{9}$$

which is an ODE describing the trajectory of a dynamical system in a learned latent space.

Suppose that the dynamical system has the form $\dot{\mathbf{x}} = f(\mathbf{x})$, and we have measurements of $\mathbf{x} = \{\mathbf{x}_1, \mathbf{x}_2, ..., \mathbf{x}_t, \mathbf{x}_{t+1}, ..., \mathbf{x}_T\}$ with time gap $\Delta t$. We first define the SINDy-class functional:

$$\mathcal{F}_{\text{SINDy}} := \{\Theta\xi : \xi \in \mathbb{R}^p\}, \tag{10}$$

where all $f \in \mathcal{F}_{\text{SINDy}}$ are functions of a convex (linear) combination of functions in $\Theta(\cdot)$, and all functions in the library are within $L^2(P)$. An example of $\Theta(x)$ is to have $[x, x^2, x^3, x^4, \sin(x), \cos(x)]$, and this could represent a dynamical system with the following form

$$\dot{f}(x) = a_1 x + a_2 x^2 + a_3 x^3 + a_4 x^4 + a_5 \sin(x) + a_6 \cos(x), \tag{11}$$

where $a_1, a_2, ..., a_6$ are constants.

Then, we consider the target function $f$ to be within $\mathcal{F}_0$.

$$\mathcal{F}_0 := \{f_0 : \sup_{x \in X} |f_0(x) - f(x)| < \epsilon, \ f \in \mathcal{F}_{\text{SINDy}}\}, \tag{12}$$

where $f_0(\cdot)$ is a function within convex (linear) combination of functions in $\Theta(\cdot)$. We further assume that $\mathcal{F}_0$ is $L$-Lipschitz.

We note that $\mathcal{F}_0$ is a wide class of functions. Since $\Theta(\cdot)$ covers polynomials and the Fourier series, the functional class $\mathcal{F}_0$ could model the governing effect for all differentiable functions from Taylor's approximation. Then, we consider the two-layer ReLU functional class $\mathcal{F}_{\text{ReLU}}$ to be

$$\mathcal{F}_{\text{ReLU}} := \{\mathbf{x} \mapsto \theta_2 \sigma(\theta_1 \mathbf{x})\}, \tag{13}$$

where $\theta_1 \in \mathbb{R}^{p \times d}$, $\theta_2 \in \mathbb{R}^{d \times p}$, and $\sigma(x) = \max(0, x)$ represents the ReLU function. In the following, we additionally require the following conditions on the input data that

$$\|\mathbf{x}\|_\infty \leq 1, \quad \|\mathbf{x}\|_2 \leq \sqrt{d}, \quad |f(\mathbf{x})| \leq \frac{1}{2}. \tag{14}$$

Define the error of dynamical system simulation as

$$\mathcal{E}(\mathbf{x}, t) = \mathbf{x}(t) - \hat{\mathbf{x}}(t). \tag{15}$$

**Theorem 1.** *Suppose we have a target function $f_0 \in \mathcal{F}_0$ and the approximation is $\hat{f}$. Suppose that the approximation error of $\hat{f}$ is up to $\epsilon$ as $\sup_x |f_0(x) - \hat{f}(x)| < \epsilon$. The generalization error is up to $E_{gen}$ with probability $1 - \delta$. Then, simulating the system up to time $T$ will reach the error with rate $\mathcal{O}\left(\frac{\epsilon + E_{gen}}{L} \exp(LT)\right)$.*

*Proof.* The error $\mathcal{E}(\cdot)$ is defined as

$$\mathcal{E}(\mathbf{x}, t) = \mathbf{x}(t) - \hat{\mathbf{x}}(t) \tag{16}$$

By taking the derivative with respect to $t$ on both sides, we get

$$\dot{\mathcal{E}}(x, t) = f_0(x) - \hat{f}(\hat{x}) \tag{17}$$
$$= [f_0(x) - f_0(\hat{x})] + \underbrace{[f_0(\hat{x}) - \hat{f}(\hat{x})]}_{\text{generalization error}} \tag{18}$$

The generalization error includes the regret and the approximation error. We first observe that $\frac{d}{dt}\|\mathcal{E}(x, t)\| \leq \|\frac{d}{dt}\mathcal{E}(x, t)\|$ from Cauchy-Schwarz. Considering the norm, due to the triangle inequality,

$$\frac{d}{dt}\|\mathbf{x}(t) - \hat{\mathbf{x}}(t)\| \leq \|f_0(\mathbf{x}) - f_0(\hat{\mathbf{x}})\| + \|f_0(\hat{\mathbf{x}}) - \hat{f}(\hat{\mathbf{x}})\| \tag{19}$$

Suppose $\hat{\mathbf{x}}$ is still within the input domain, from the prior result, we know with probability $1 - \delta$ we have

$$\sup_{\mathbf{x}} \|f(\mathbf{x}) - \hat{f}(\mathbf{x})\| \leq \epsilon + E_{\text{gen}}. \tag{20}$$

Then, since the functional class $\mathcal{F}_0$ is $L$-Lipschitz, we have

$$\frac{d}{dt} \|\mathbf{x}(t) - \hat{\mathbf{x}}(t)\| \leq L \|\mathbf{x} - \hat{\mathbf{x}}\| + \epsilon + E_{\text{gen}}. \tag{21}$$

By applying the differential form of Grönwall's inequality, we see

$$\mathcal{E}(x, t) \leq \frac{\epsilon + E_{\text{gen}}}{L} \left(e^{LT} - 1\right), \tag{22}$$

which is the upper bound of the error. $\qquad\square$

By observing Thm. 1, we found the error has rate $\mathcal{O}\left(\frac{\epsilon + E_{\text{gen}}}{L} e^{LT}\right)$. Therefore, as $T$ increases, the error will be magnified with a factor of $e^T$. This type of error accumulation will create problems for neural networks. By increasing the power of approximation (having a smaller $\epsilon$), the neural network will lose the ability to generalize, leading to a large $E_{\text{gen}}$. Meanwhile, when $E_{\text{gen}}$ is large, the extrapolation may be out of the input domain $\mathcal{X}$, resulting in unstable extrapolations of the dynamical system. However, the SINDy-class functional is inherently designed to avoid this issue. We analyze the error for the SINDy-class functional in the following.

**Theorem 2.** *The SINDy-class functional, with probability $1 - \delta$, will have an error with order*

$$\mathcal{E}(T) = \mathcal{O}\left(\frac{\epsilon + \sqrt{\frac{M_n}{n}}}{L} e^{LT}\right) \tag{23}$$

*Proof.* First, we note that by definition, the optimal solution $\xi_0$ can reach an error of $\epsilon$ pointwise. Then, we consider the empirical risk minimizer $\hat{\xi}$, and study the regret:

$$\text{Reg}(\hat{\xi}) = P\ell(\hat{\xi}) - P\ell(\xi_0). \tag{24}$$

In the task of regression, we know this $\ell(\cdot)$ is

$$\ell(x, \xi) = [f_0(x) - \Theta(x)\xi]^2 \tag{25}$$

Define $g \in \mathcal{G} := \{x \mapsto \ell(x, \xi) : \xi \in \Xi\}$. From the empirical process theory, we know

$$\text{Reg}(\hat{\xi}) \leq P\ell(\hat{\xi}) - P\ell(\xi_0) + \left[P_n\ell(\xi_0) - P_n\ell(\hat{\xi})\right] \tag{26}$$

$$= (P_n - P)[\ell(\xi_0) - \ell(\hat{\xi})] \tag{27}$$

$$\leq |(P_n - P)\ell(\xi_0)| + (P_n - P)\ell(\hat{\xi})| \tag{28}$$

$$\leq 2 \sup_{\xi \in \Xi} |(P_n - P)\ell(\xi)| = 2 \sup_{g \in \mathcal{G}} |(P_n - P)g| = 2 \|P_n - P\|_{\mathcal{G}} \tag{29}$$

So we need to study this functional class $\mathcal{G}$.

We expand the definition of $\mathcal{G} := \{x \mapsto [f_0(x) - \Theta(x)\xi]^2 : \xi \in \Xi\}$. Using the definition of $f_0$, we have $\mathcal{G} := \{x \mapsto [\Theta(x)(\xi_0 - \xi) + \eta(x)]^2 : \xi \in \Xi\}$ where $\eta(x)$ denotes some quantity that is upper bounded by $\eta > 0$.

We see from Markov's inequality that for $t > 0$ and some $a > 0$,

$$\mathbb{P}\left(\text{Reg}(\hat{\xi}) > \left(\frac{M_n}{n}\right)^a t\right) \leq \mathbb{P}\left(2 \|P_n - P\|_{\mathcal{G}} > \left(\frac{M_n}{n}\right)^a t\right) \tag{30}$$

$$\leq \frac{2\mathbb{E}\|P_n - P\|_{\mathcal{G}}}{\left(\frac{M_n}{n}\right)^a t} \tag{31}$$

$$\leq \frac{2\mathbb{E}\|R_n\|_{\mathcal{G}}}{\left(\frac{M_n}{n}\right)^a t}, \tag{32}$$

where $\mathbb{E}\|R_n\|_{\mathcal{G}}$ denotes the Rademacher complexity. $R_n(g) = \frac{1}{n}\sum_{i=1}^{n}\epsilon_i g(x_i)$; $\epsilon_i$'s are independent Rademacher random variables $\text{Unif}(0, 1)$, and $\|R_n\|_{\mathcal{G}} = \sup_{g \in \mathcal{G}} |R_n(g)|$.

We start from the case that $x$ is univariate. We assume that the data we have have the form $\{(\mathbf{x}_i, \mathbf{y}_i)\}_{i=1}^{N}$ where $\mathbf{x}_i$ has a bounded domain and $\mathbf{y}_i$ is within range $[-\frac{1}{2}, \frac{1}{2}]$. Define the empirical risk minimizer $\xi_0 : w \mapsto \mathbb{E}_P[\mathbf{y} \mid \mathbf{x} = \mathbf{x}_i]$. The estimator $\hat{\xi}_n$ minimizes empirical MSE $\mathcal{R}_n : \xi \mapsto \frac{1}{n}\sum_{i=1}^{n}[\mathbf{y}_i - \xi(\mathbf{x}_i)]^2$. We denote $\Theta_M$ be the class of functions whose range falls in $[-\frac{1}{2}, \frac{1}{2}]$ with total variation not larger than $M$. The total variation is defined as a class of functions $h : [0, 1] \mapsto \mathbb{R}$ that satisfies $\forall k \in \mathbb{N}$, and $\forall 0 = x_0 < x_1 < ... < x_{n-1} < x_k < 1$:

$$\sum_{i=1}^{n-1} |h(x_{i+1}) - h(x_i)| \leq M. \tag{33}$$

By plugging in the definition of $\mathcal{G}$, we find

$$\sum_{i=0}^{n-1} |g(x_{i+1}; f_0, \xi) - g(x_i; f_0, \xi)| \tag{34}$$

$$= \sum_{i=0}^{n-1} |(\Theta(x_{i+1})(\xi_0 - \xi) + \epsilon(x_{i+1}))^2 - (\Theta(x_i)(\xi_0 - \xi) + \epsilon(x_i))^2| \tag{35}$$

$$= \sum_{i=0}^{n-1} |(\xi - \xi_0)^2[\Theta(x_{i+1})^2 - \Theta(x_i)^2] + 2(\xi_0 - \xi)[\Theta(x_{i+1})\eta(x_{i+1}) - \Theta(x_i)\eta(x_i)] + [\eta(x_{i_1})^2 - \eta(x_i)^2]| \tag{36}$$

We see that the above quantity is bounded because $\xi$ has a bounded domain, $\Theta(\cdot)$ is Lipschitz within $[0, 1]$, and $\epsilon(\cdot)$ is bounded by a constant. We note this constant as $M_n$, which corresponds to $M_n$ in Eq. 30.

Say $Q$ is a continuous distribution on $[0, 1]$. The covering number of the functional class $\mathcal{G}$ and the covering number of $\Theta_M$ is connected through $\mathcal{N}(\epsilon, \mathcal{G}, L^2) \leq \mathcal{N}(\epsilon, \Theta_M, L^2)$.

Suppose that we have an $\epsilon/4$-cover of $\theta_M$ which contains $\{\theta_1, \theta_2, ..., \theta_n\}$. We see that $\{\ell(\theta_j)\}$ is an $\epsilon$-cover of $\mathcal{G}$ by

$$\|\ell(\theta) - \ell(\theta_j)\|_{\mathcal{L}^2(Q)}^2 = \int [(y - \theta x)^2 - (y - \theta_j x)^2]^2 dQ(x) \tag{37}$$

$$= \int [(2y - \theta x - \theta_j x)(\theta x - \theta_j x)]^2 dQ(x) \tag{38}$$

$$\leq \sup_{x,y}(2y - \theta x - \theta_j x)^2 \int (\theta x - \theta_j x)^2 dQ(x) \tag{39}$$

$$[\{\theta_j\} \text{ is an } \frac{\epsilon}{4}\text{-cover}] \leq \epsilon^2. \tag{40}$$

Then, from Lemma 1, we know that $\mathcal{N}(\epsilon, \Theta_M, \mathcal{L}^2)$ is bounded by $\frac{13}{\epsilon}$. Since the Rademacher random variable is sub-Gaussian, we can control the Rademacher complexity further using the Dudley's entropy integral,

$$\mathbb{E}\|R_n\|_{\mathcal{G}} \leq 8n^{-1/2} \sup_Q \int_0^{\infty} \sqrt{\log \mathcal{N}(\epsilon, \Theta_{M_n}, \mathcal{L}^2(Q))} d\epsilon \tag{41}$$

$$[D := \sup_{x,y} d(x, y)] = 8n^{-1/2} \sup_Q \int_0^{D} \sqrt{\log \mathcal{N}(\epsilon, \Theta_{M_n}, \mathcal{L}^2(Q))} d\epsilon \tag{42}$$

$$\leq 8n^{-1/2} \sup_Q \int_0^{D} \sqrt{\log \mathcal{N}(\epsilon, \Theta_{M_n}, \mathcal{L}^2(Q))} d\epsilon \tag{43}$$

$$\leq \frac{CM_n}{\sqrt{n}}. \tag{44}$$

From a similar process, we can bound the multivariate input version by some other constant $C_p$ that bounds the norm of the possible parameters (i.e. $\|\xi\|_2 \leq C_p$). By applying the differential form of Grönwall's inequality, we see

$$\mathcal{E}(x,t) \leq \left(\frac{2CM_n}{L\sqrt{n}} + \epsilon\right)\left(e^{LT} - 1\right), \tag{45}$$

which is the upper bound of the error.

We note here that in theory it is possible to push $\mathcal{E}(x,t) = O_P\left(n^{-2/3}\right)$ under the condition $M_n = o(n)$, which is better than the current rate $O_P\left(n^{-1/2}\right)$. $\qquad\square$

Here, we find that the generalization error is independent of $\epsilon$ for the SINDy-class functional. Therefore, the approximation-generalization tradeoff does not hurt SINDy when the target function is within $\mathcal{F}_0$.

Now, we see the proof for neural networks. There are two factors that might cause the generalization to be large. First, when the target functional to learn is very close to $\mathcal{F}_{\text{SINDy}}$ with small $\epsilon$. In this case, to obtain a better approximation, the network has to increase the size of the hidden layer. This will lead to an increase in terms of the Rademacher complexity and therefore the generalization error. In addition, the input data for neural networks are different. Unlike the ODE forward integration structure in SINDy, neural networks frequently use the previous $L$ data points as input to predict the value at $t_{L+1}$. This will lead to a larger input space, which will also increase the generalization error. We show the details of the theorem in the following.

**Theorem 3.** *If we use a function in $\mathcal{F}_{ReLU}$ to learn a dynamical system, we expect to have an error of up to*

$$\mathcal{E}(T) = \mathcal{O}\left(\epsilon e^{LT} + \frac{W_1 W_2}{\sqrt{n}}\sqrt{\log\left(\frac{\Lambda(d)}{\epsilon}\right)}e^{LT}\right), \tag{46}$$

*where $W_1, W_2, \Lambda(d)$ satisfies $\|\theta_1\|_2 \leq W_1$, $\|\theta_2\|_2 \leq W_2$, and $\int_{\mathbb{R}^d}\|\omega\|_2 |\hat{f}(\omega)|d\omega \leq \Lambda(d)$, where $\int \|\omega\|_2 |\hat{f}(\omega)|d\omega$.*

*Proof.* We know that to reach an approximation error of $\epsilon$ uniformly, from Barron's approximation theorem, we know:

$$\inf_{\hat{f}\in\mathcal{F}_{\text{ReLU}}} \sup_x |f_0(x) - \hat{f}(x)| \leq \frac{\Lambda(d)}{p}, \tag{47}$$

where $\Lambda(d)$ is the Barron constant and $p$ is the size of hidden layer. We know, under the same $\Lambda(d), W_1, W_2$, we need $p \geq \left(\frac{\Lambda(d)}{\epsilon}\right)^2 = \mathcal{O}\left(\frac{\Lambda(d)^2}{\epsilon^2}\right)$.

We can know the Radamacher complexity from classical result (Bartlett & Mendelson, 2002) (c.f. Thm. 18) is bounded by

$$\hat{\mathcal{R}}_n(\mathcal{F}) \leq \frac{CW_1 W_2\sqrt{\log(p)}}{\sqrt{n}}, \tag{48}$$

for some constant $C$.

From the required precision for approximation, we know that the generalization error scales with

$$E_{\text{gen}} = \mathcal{O}\left(\frac{W_1 W_2}{\sqrt{n}}\sqrt{\frac{\Lambda(d)}{\epsilon}}\right). \tag{49}$$

Using the Grönwall's inequality again, we see

$$\mathcal{E}(T) = \mathcal{O}\left(\epsilon e^{LT} + \frac{W_1 W_2}{\sqrt{n}}\sqrt{\log\left(\frac{\Lambda(d)}{\epsilon}\right)}e^{LT}\right). \tag{50}$$

$\qquad\square$

From Thm. 3, we see the dilemma of neural networks. The generalization error will increase as $\epsilon$ gets smaller. In this case, when $f_0$ can be closely represented via $\mathcal{F}_{\text{SINDy}}$ with $\epsilon \to 0$, the neural network will fail to perform reasonable extrapolations. In practice, regularization techniques can be applied to mitigate unstable predictions. However, this will reduce the accuracy of the approximation and make the theoretical behavior difficult to characterize.

Another way is to show the bound using (Bartlett & Mendelson, 2002) will incorporate a factor of $\mathcal{O}\left(\sqrt{d}\right)$. It is also possible to show the instability of neural networks due to the magnification of the input space by the factor of $L$ (length of the input trajectory).

### B.1 TECHNICAL LEMMAS

**Lemma 1.** *(Bartlett et al., 1997). For all $\epsilon < \frac{1}{12}$, we have*

$$(\log_2 e)\frac{1}{54\epsilon} \leq \log_2 \mathcal{N}(\epsilon, \mathcal{F}_1, \mathcal{L}^2) \leq \frac{13}{\epsilon}, \tag{51}$$

*where $\mathcal{F}_1$ has bounded variation and takes value from $[0,1]$ to $[-1/2, 1/2]$. $\mathcal{L}^2$ denotes the norm $||f||_{\mathcal{L}^1(P)} = \int |f(x)|dP(x)$.*

**Definition 1.** *A function $g : S \times S \mapsto [0, \infty)$ is called a pseudometric on $S$ if:*

*(a) $d(x,x) = 0, \; \forall x \in S$.*

*(b) $d(x,y) = d(y,x), \; \forall x, y \in S$.*

*(c) $d(x,z) \leq d(x,y) + d(y,z), \; \forall x, y, z \in S$*

We denote a pseudometric space by $(S, d)$.

**Definition 2.** *Let $(S,d)$ be a pseudometric space and let $T \subseteq S$. $T_1 \subseteq T$ is called an $\epsilon$-cover if $\forall \theta \in T$, there is a $\theta_1 \in T_1$ s.t. $d(\theta, \theta_1) \leq \epsilon$.*

*And the $\epsilon$-covering number of $T$ is defined as*

$$\mathcal{N}(\epsilon, T, d) = \min\{|T_1| : T_1 \text{ is an } \epsilon\text{-cover of } T\}, \tag{52}$$

*where $|T_1|$ denotes the cardinality of set $T_1$.*

**Lemma 2.** *(Barron, 1993) To reach an approximation error of $\epsilon$, fixing the norm of network parameters, we need to satisfy*

$$p \leq \frac{C \|f\|_{\mathcal{B}}^2 W^2}{\epsilon^2}. \tag{53}$$

## C EXPERIMENTAL DETAILS

### C.1 SEA-SURFACE TEMPERATURE DATA

For the SST data in SINDy-SHRED, we set the latent dimension to 3 because we observe only minor impacts on the reconstruction accuracy when the latent dimension is $\geq 4$. We include 2 stacked GRU layers and consider the , and a two-layer ReLU decoder with 350 and 400 neurons. For the E-SINDy regularization, we set the polynomial order to be 3 and the ensemble number is 10. In the latent hidden-state forward simulation, we use Euler integration with $dt = \frac{1}{520}$, which will generate the prediction of next week via 10 forward integration steps. During training, we apply the AdamW optimizer with a learning rate of $1e^{-3}$ and a weight decay of $1e^{-2}$. The batch size is 128 with 1,000 training epochs. The thresholds for E-SINDy range uniformly from 0.1 to 1.0, and the thresholding procedure will be executed every 100 epochs. We use dropout to avoid overfitting with a dropout rate of 0.1. The training time is within 30 minutes from a single NVIDIA GeForce RTX 2080 Ti.

### C.2 3D ATMOSPHERIC OZONE CONCENTRATION

For the ozone data, we set the lag parameter is set to 100. Thus, for each input-output pair, the input consists of the 62.5 day measurements of the selected sensors, while the output is the measurement

across the entire 3D domain. In SINDy-SHRED, we follow the same network architecture as in the SST experiment. We set $dt = 0.025$, and the thresholds for E-SINDy range uniformly from 0.015 to 0.15. The thresholding procedure will be executed every 300 epochs, and we apply AdamW optimizer with learning rate $1e^{-3}$.

### C.3 FLOW OVER A CYLINDER

In the flow over a cylinder experiment, we follow the same settings as in the prior experiments and select the latent dimension to be 4. The forward integration time step is set to $dt = \frac{1}{300}$ corresponding to the frame rate of 30 FPS. We set the batch size at 64 and the learning rate to $5e^{-4}$. The thresholding procedure is executed every 300 epochs with thresholds ranging from $(1e^{-4}, 1e^{-3})$.

### C.4 BASELINE EXPERIMENT ON PENDULUM

**Autoregressive training.** The raw pendulum data are collected from a 14-second GoPro recording. The raw data are present difficulties during training because of their high-dimensionality ($1080 \times 960$), so we follow the same preprocessing procedure as in (Mars Gao & Nathan Kutz, 2024) to obtain a set of training data with 390 samples, width 24 and height 27. For most of the models, we apply autoregressive training to help the model achieve better long-term prediction capabilities. From the initial input $\{X_1, X_2, ..., X_L\}$ with lag $L$, the model autoregressively predicts the next frame $\hat{X}_{L+1}$ and use it as a new input $\{X_2, X_3, ..., \hat{X}_{L+1}\}$. This step will be repeated $L$ times to obtain $\{\hat{X}_{L+1}, \hat{X}_{L+2}, ..., \hat{X}_{2L}\}$. We treat this as the prediction and optimize the loss from this quantity. In the following baseline models, we uniformly set $L = 20$.

#### C.4.1 BASELINE METHODS AND SINDy-SHRED SETTING

**ResNet.** We use the residual neural network (ResNet) (He et al., 2016) as a standard baseline. We set the input sequence length to 20, and we predict the next frames autoregressively. For ResNet, the first convolutional layer has 64 channels with kernel size 3, stride 1 and padding 1. Then, we repeat the residual block three times with two convolutional layers. We use ReLU as the activation function. After the residual blocks, the output is generated via a convolutional layer with kernel size 1, stride 1, and padding 0. We set the batch size to 8, and we use AdamW optimizer with learning rate $1e^{-3}$, weight decay $1e^{-2}$ for the training of 500 epochs.

**SimVP.** SimVP (Gao et al., 2022) is the recent state-of-the-art method for video prediction. This method utilizes ConvNormReLU blocks with a spatio-temporal features translator (i.e. CNN). The ConvNormReLU block has two convolutional layers with kernel size 3, stride 1, and padding 1. After 2D batch normalization and ReLU activation, the final forward pass includes a skip connection unit before output. The encoder first performs a 2D convolution with 2D batch normalization and ReLU activation. Then, three ConvNormReLU blocks will complete the input sequence encoding process. The translator in our implementation is a simple CNN which contains two convolutional layers. The decoder has a similar structure to the encoder by reversing its structure. We similarly set the batch size to 8 with AdamW optimizer for 500 epochs.

**ConvLSTM.** Convolutional Long Short-Term Memory (Shi et al., 2015) is a classical baseline for the prediction of video sequence and scientific data (e.g. weather, radar echo, and air quality). The ConvLSTM utilizes features after convolution and performs LSTM modeling on hidden states. The ConvLSTM model has two ConvLSTM cells that have an input 2D convolutional layer with kernel size 3 and padding 1 before the LSTM forward pass. The decoder is a simple 2D convolution with kernel size 1, and zero padding. We similarly set the batch size to 8 with AdamW optimizer for 500 epochs.

**PredRNN.** PredRNN (Wang et al., 2017) is a recent spatiotemporal modeling technique that builds on the idea of ConvLSTM. We follow the same network architecture setting as in ConvLSTM and similarly set the batch size to 8 with AdamW optimizer for 500 epochs.

**SINDy-SHRED.** We select and fix 100 pixels as sensor measurements from the entire 648 dimensional space. We remove non-informative sensors, defined as remaining constant through the entire

video. We set the lag to 60. For the setting of network architecture in SINDy-SHRED, we follow the same settings as in the prior experiments but with latent dimension of 1. The timestep of forward integration is set to $dt = \frac{1}{300}$ corresponding to frame rate of the video at 30 FPS. We set the batch size at 8 and the learning rate to $5e^{-4}$. The thresholding procedure is executed every 300 epochs with thresholds ranging from $(0.4, 4.0)$. We include 3 stacked GRU layers, and a two-layer ReLU decoder with 16 and 64 neurons. We use dropout to avoid overfitting with a dropout rate of 0.1. SINDy-SHRED discovers two candidate models.

## D    EXPERIMENT ON THE 2D KOLMOGOROV FLOW

The 2D Kolmogorov flow data is a chaotic turbulent flow generated from the pseudospectral Kolmogorov flow solver (Canuto et al., 2007). The solver numerically solves the divergence-free Navier-Stokes equation:

$$\begin{cases} \nabla \cdot \boldsymbol{u} = 0 \\ \partial_t \boldsymbol{u} + \boldsymbol{e}\nabla \boldsymbol{u} = -\nabla \boldsymbol{p} + \mathbf{v}\Delta \boldsymbol{u} + f \end{cases} , \tag{54}$$

where $\boldsymbol{u}$ stands for the velocity field, $\boldsymbol{p}$ stands for the pressure, and $f$ describes an external forcing term. Setting the Reynolds number to 30, the spatial field has resolution $80 \times 80$. We simulate the system forward for 180 seconds with $6,000$ available frames. We standardize the data within the range of $(0, 1)$ and randomly fix 10 sensors from the 6,400 available spatial locations (0.16%). The lag parameter is set to 360.

For the setting of SINDy-SHRED, we slightly change the neural network setting because the output domain is 2D. Therefore, after the GRU unit, we use two shallow decoders to predict the output of the 2D field. The two decoders are two-layer ReLU networks with 350 and 400 neurons. We set the latent dimension to 3. The time step for forward integration is set to $dt = 0.003$ which corresponds to the FPS during data generation. We set the batch size to 256 and the learning rate to $5e^{-4}$ using the Lion optimizer (Chen et al., 2024). The thresholding procedure is executed every 100 epoch with the total number of training epochs as 200. The thresholds range from $(0.4, 4)$.

As a chaotic system, the latent space of the Kolmogorov flow is much more complex than all the prior examples we considered. Thus, we further apply seasonal-trend decomposition from the original latent space. We define the representation of the latent hidden state space after decomposition as $(z_1, z_2, z_3, z_4, z_5, z_6)$, where $(z_{2i}, z_{2i+1})$ is the seasonal trend pair of the original latent space.

$$\begin{cases} \dot{z}_1 &= -0.007z_3 + 0.009z_5, \\ \dot{z}_2 &= -0.207z_4, \\ \dot{z}_3 &= -0.011z_1 - 0.008z_5, \\ \dot{z}_4 &= 0.103z_2, \\ \dot{z}_5 &= -0.012z_1 + 0.006z_3. \\ \dot{z}_6 &= 0.151z_1z_2. \end{cases} \tag{55}$$

In Eqn. 55, we find that $z_1, z_3, z_5$ are essentially a linear system. $z_2, z_4, z_6$ capture higher-order effects that are difficult to model without signal separation. We generate the trajectory from the initial condition at time point 0 and perform forward integration in Fig. 12. As we increase the Reynolds number, the discovery fails to produce robust predictions.

This representation also demonstrates nice predictions for future frames. In Fig. 13, the future prediction has an averaged MSE error of 0.035 for all available data samples. The sensor-level prediction in Fig. 23 further demonstrates the details of the reconstruction.

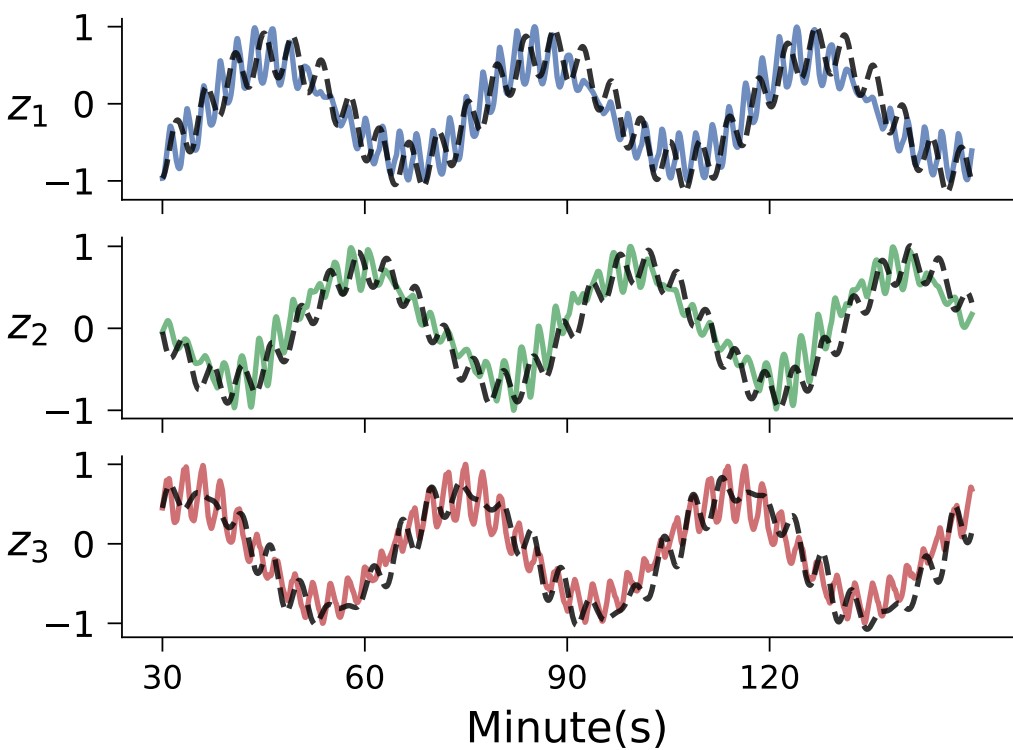

Figure 12: Extrapolation of latent representation in SINDy-SHRED from the discovered dynamical system for the 2D Kolmogorov flow data. Colored: true latent representation. Black: SINDy extrapolation.

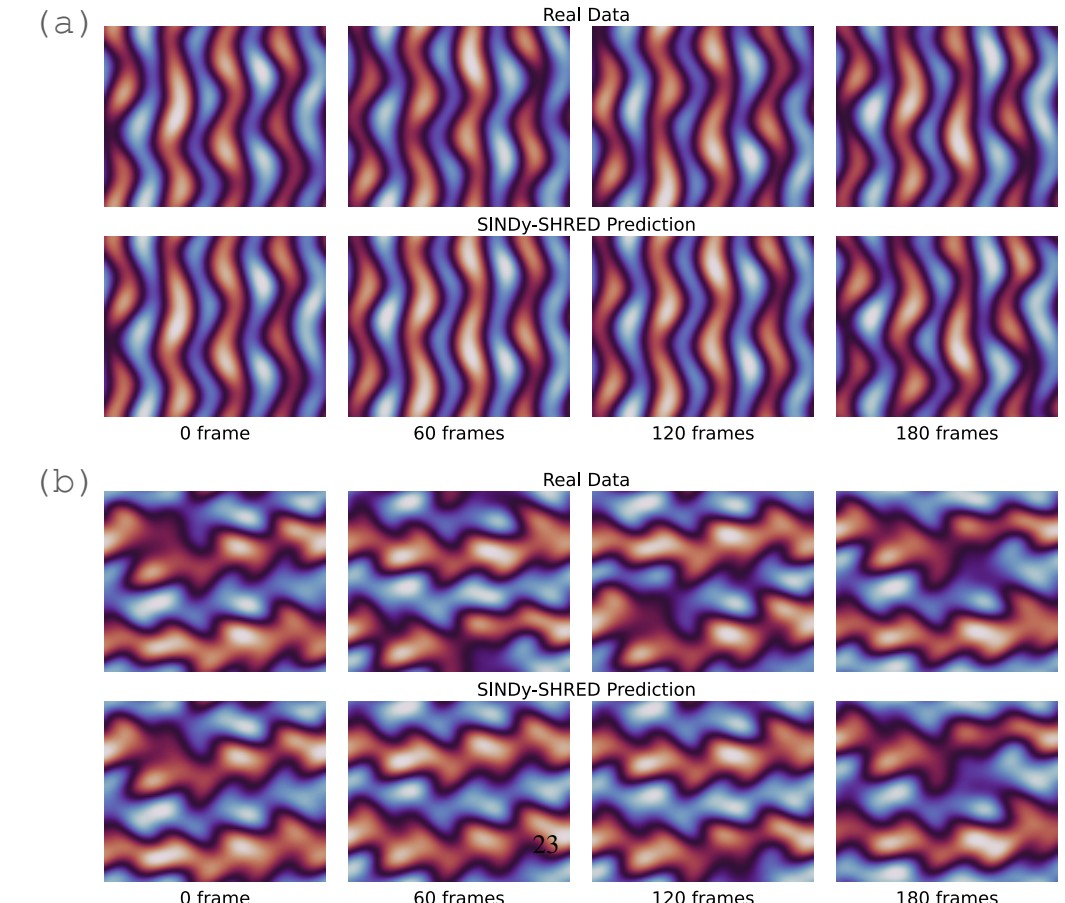

# E   SENSOR LEVEL PLOTS OF EXPERIMENTS

## E.1   SEA SURFACE TEMPERATURE

**3D visualization of SINDy-SHRED**

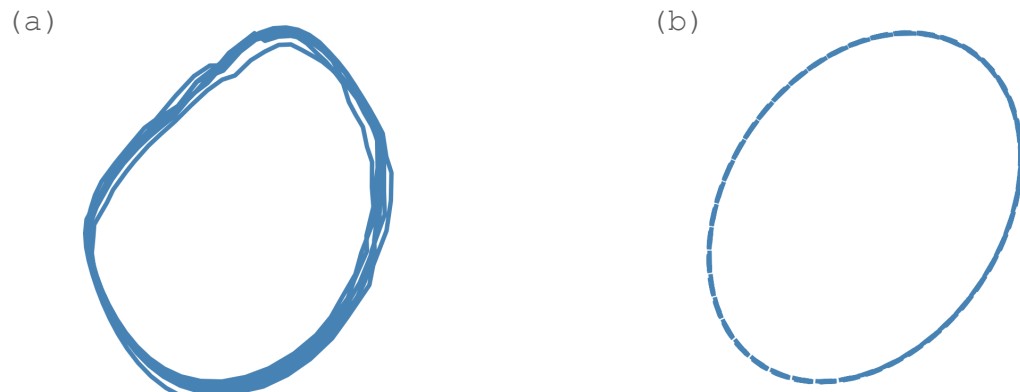

Figure 14: 3D reconstruction of the original latent space and SINDy simulated latent space.

**Long-term extrapolation of SINDy-SHRED.**

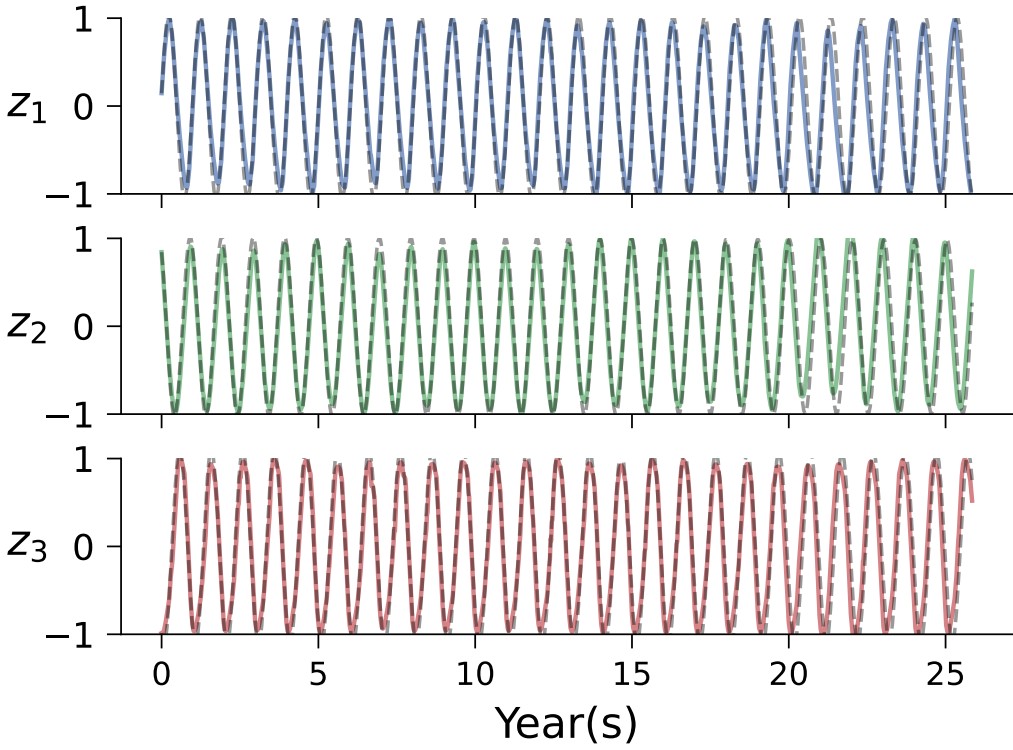

Figure 15: Extrapolation of latent representation in SINDy-SHRED from the discovered dynamical system for SST over the entire 27 years. Colored: true latent representation. Grey: SINDy extrapolation.

**Sensor-level prediction on the SST dataset.**

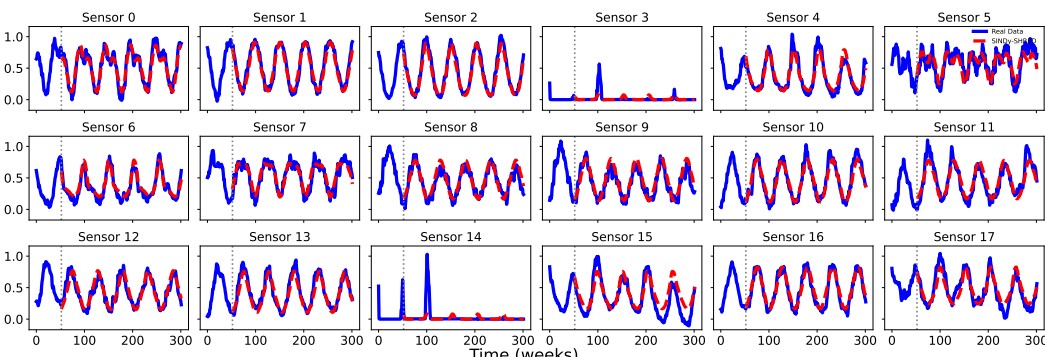

Figure 16: Extrapolation of SINDy-SHRED for sensor-level predictions on the SST data. We randomly picked 18 sensors from spatial locations that are not in the sparse sensor training. The extrapolation shows the SINDy-SHRED prediction for the following 300 weeks.

### E.2 OZONE DATA

**Convergence behavior of SINDy-SHRED on the Ozone dataset.**

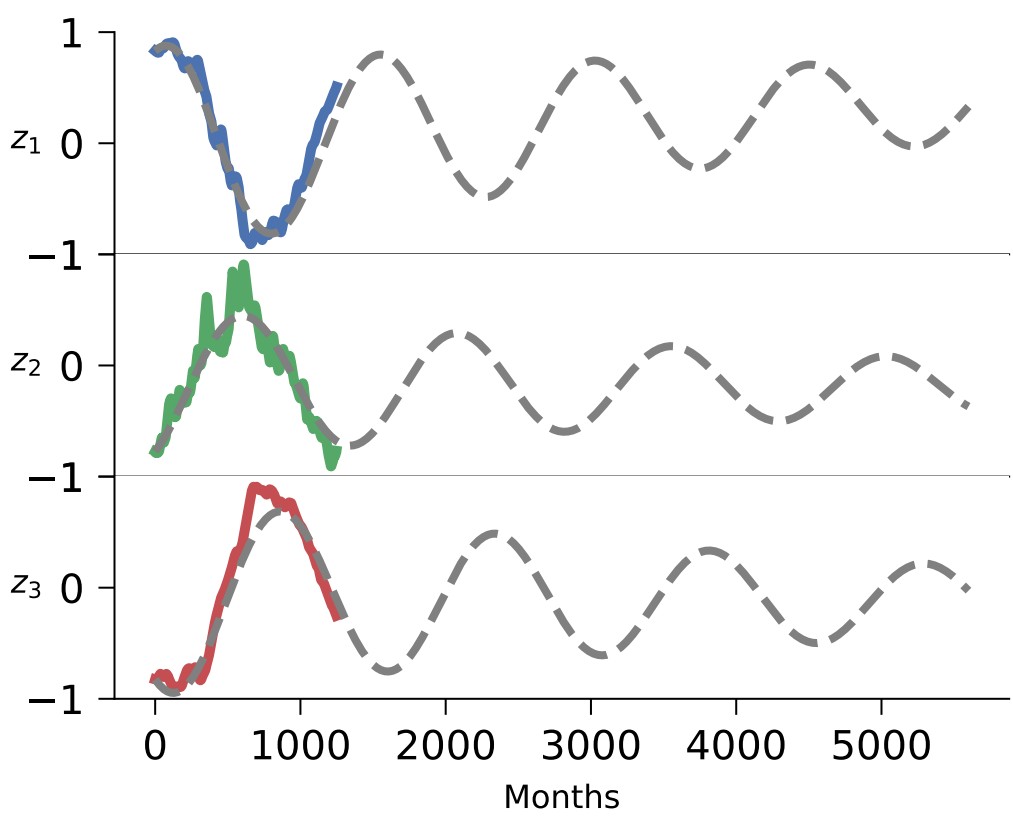

Figure 17: Long term extrapolation of Ozone data. The latent SINDy model presents a convergence behavior towards the mean-field solution.

**Sensor-level prediction on the Ozone dataset.**

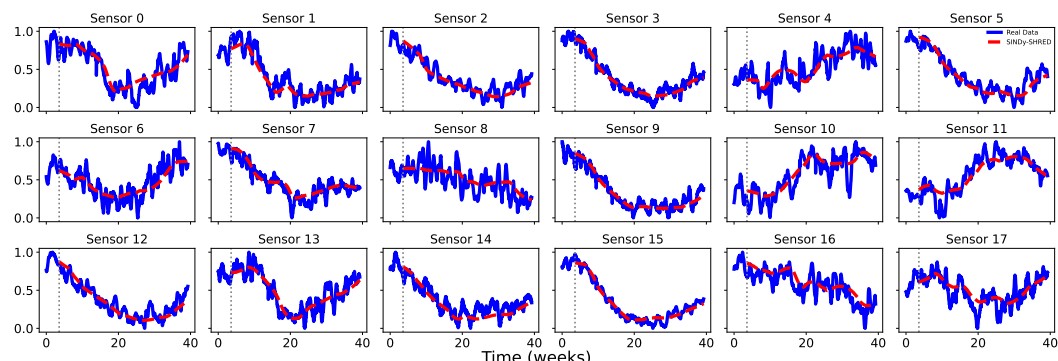

Figure 18: Extrapolation of SINDy-SHRED for sensor-level predictions on the Ozone data. We randomly picked 18 sensors from spatial locations that are not in the sparse sensor training. The extrapolation shows the SINDy-SHRED prediction for the following 40 weeks.

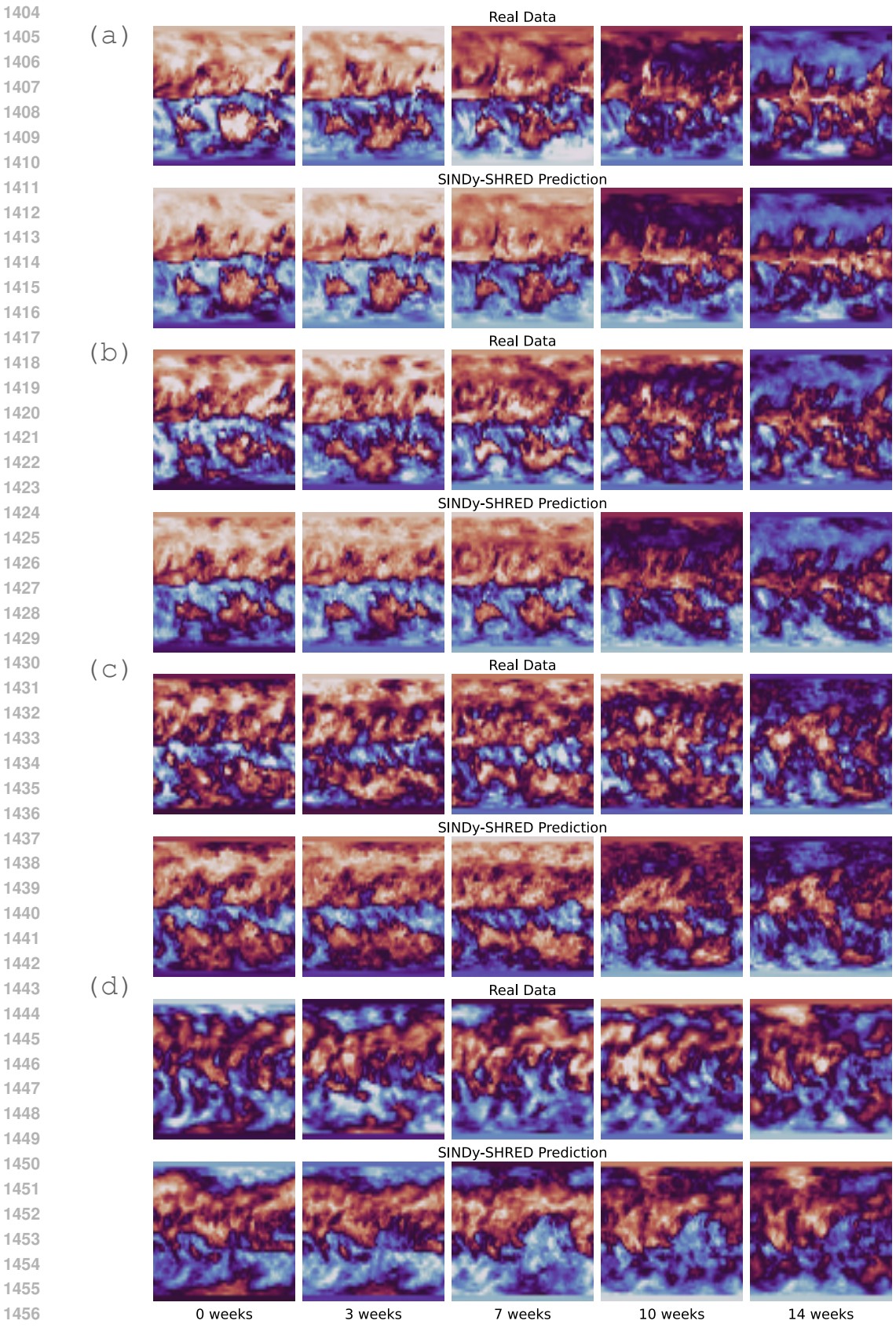

Figure 19: Reconstruction of atmospheric ozone concentration data for different elevation (a) 0 km (b) 4 km (c) 8 km (d) 12 km.

### E.3 FLOW OVER A CYLINDER

**Sensor-level prediction on the flow over a cylinder dataset.**

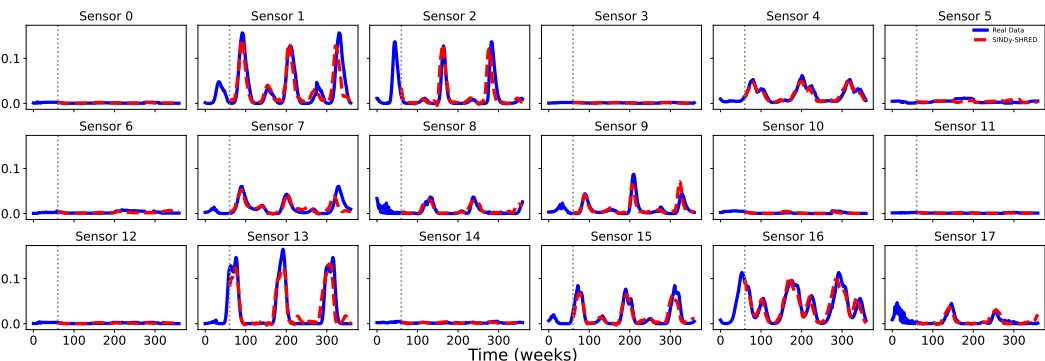

Figure 20: Extrapolation of SINDy-SHRED for sensor-level predictions on the flow over a cylinder data. We randomly picked 18 sensors from spatial locations that are not in the sparse sensor training. The extrapolation shows the SINDy-SHRED prediction for the following 400 frames.

**Long-term extrapolation on the flow over a cylinder dataset.**

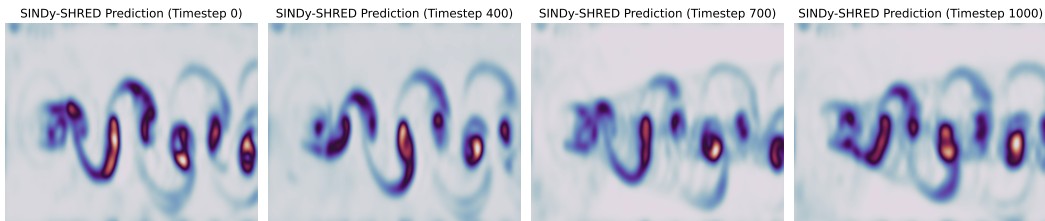

Figure 21: Prediction of the flow over a cylinder data from time step 0 (reconstruction) to 1000 frames. We note this extrapolation is completely out of the dataset. The real data for testing is only available up to 500 frames.

### E.4 PENDULUM

**Sensor-level prediction on the moving pendulum dataset.**

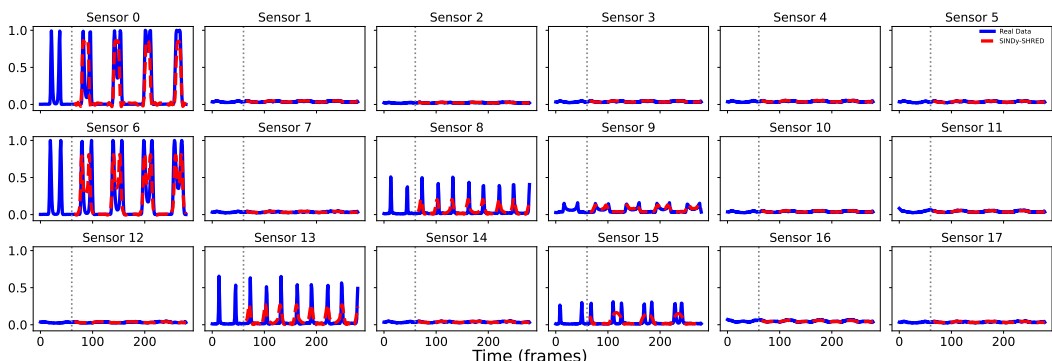

Figure 22: Extrapolation of SINDy-SHRED for sensor-level predictions on the moving pendulum data. We randomly picked 18 sensors from spatial locations that are not in the sparse sensor training. The extrapolation shows the SINDy-SHRED prediction for the following 382 frames.

## E.5 KOLMOGOROV FLOW

**Sensor-level prediction on the chaotic 2D Kolmogorov flow dataset.**

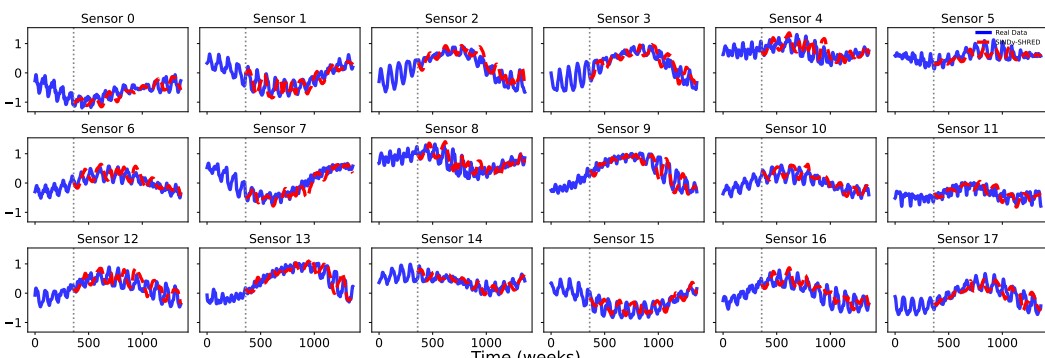

Figure 23: Extrapolation of SINDy-SHRED for sensor-level predictions on the 2D Kolmogorov flow data. We randomly picked 18 sensors from spatial locations that are not in the sparse sensor training. The extrapolation shows the SINDy-SHRED prediction for the following 1500 frames.