# OpenReview forum: "Sparse identification of nonlinear dynamics with Shallow Recurrent Decoder Networks"
_ICLR.cc/2025/Conference — ICLR 2025 Conference Withdrawn Submission_

### Official Review · Reviewer_sEDc · 2024-10-25

**Soundness:** 2
**Presentation:** 3
**Contribution:** 1
**Rating:** 3
**Confidence:** 4

**Summary:**

The paper introduces a modified SHallow REcurrent Decoder (SHRED) framework with the addition of Sparse Identification of Nonlinear Dynamics (SINDy) regularization within the recurrent component. The novelty of the work is quite limited, as it primarily builds upon the existing SHRED network architecture with minimal modifications, focusing on the recurrent aspect by substituting LSTMs with GRUs. This incremental adjustment lacks substantial theoretical or experimental justification, which diminishes the paper’s impact.

**Strengths:**

- The SINDy regularization within the GRU-based recurrent component adds a degree of interpretability to the latent space, though this claim lacks rigorous experimental support.
- There is some empirical evidence that SINDy-SHRED performs efficiently on spatio-temporal reconstruction tasks, but this is limited to specific datasets and lacks broader validation.

**Weaknesses:**

- The model is largely based on the original SHRED framework, with minimal changes restricted to the recurrent component. An ablation study showing the comparative effectiveness of GRUs over LSTMs in creating smoother latent spaces is missing. Additionally, the roles of SINDy regularization and choice of recurrent unit remain unclear without component-wise analysis.
- Surprisingly, the paper does not include the original SHRED network in its baselines. Without this comparison, the improvements claimed by SINDy-SHRED are hard to evaluate objectively, especially considering the minimal nature of the modifications.
- The comparison with other baselines is restricted to the last dataset (pendulum recording) instead of across all datasets. Further, Table 1 highlights an odd pattern: SINDy-SHRED performs worse in short-term forecasting but somehow improves in longer leads, a trend that is not convincingly explained. This might be helpful for establishing a more rigorous baseline comparison in the future [1].
- The claim on page 4, line 182, regarding spectral bias in reduced-order models is misleading. The cited works, SHRED, and even the proposed SINDy-SHRED itself do not directly tackle spectral bias. This issue, inherent to MLP architectures, would require changes in network structure or activation types to address effectively [2-4]. The current model does not provide a stronger representation in terms of combating spectral bias as claimed.

1. Tan, C., Li, S., Gao, Z., Guan, W., Wang, Z., Liu, Z., Wu, L. and Li, S.Z., 2023. Openstl: A comprehensive benchmark of spatio-temporal predictive learning. Advances in Neural Information Processing Systems, 36, pp.69819-69831.
2. Rahaman, N., Baratin, A., Arpit, D., Draxler, F., Lin, M., Hamprecht, F., Bengio, Y. and Courville, A., 2019, May. On the spectral bias of neural networks. In International conference on machine learning (pp. 5301-5310). PMLR.
3. Tancik, M., Srinivasan, P., Mildenhall, B., Fridovich-Keil, S., Raghavan, N., Singhal, U., Ramamoorthi, R., Barron, J. and Ng, R., 2020. Fourier features let networks learn high frequency functions in low dimensional domains. Advances in neural information processing systems, 33, pp.7537-7547.
4. Sitzmann, V., Martel, J., Bergman, A., Lindell, D. and Wetzstein, G., 2020. Implicit neural representations with periodic activation functions. Advances in neural information processing systems, 33, pp.7462-7473.

**Questions:**

Overall, the limited originality, lack of critical experiments, and reliance on vague claims make this paper feel incomplete. Including ablation studies, clarifying comparative methods, and tightening claims around spectral bias would be essential steps to make the contribution more compelling. See the Weaknesses for detailed suggestions.

---

### Official Review · Reviewer_55op · 2024-10-28

**Soundness:** 3
**Presentation:** 3
**Contribution:** 3
**Rating:** 6
**Confidence:** 5

**Summary:**

This paper proposes SINDy-SHRED, which is a hybrid network that combines deep neural networks and physics discovery. The pipeline uses sampling methods to obtain low-dimensional sensor measurements, GRUs for learning low-dim dynamics, SINDy for incorporating physics, and a shallow decoder for reconstruction. The proposed method is flexible, easy, efficient, and robots. This method has demonstrated effectiveness across many synthetic and real-world datasets.

**Strengths:**

- The paper is interesting due to the incorporation of potential physics into the machine learning models for learning long-term dynamics.

- The proposed method shows good flexibility, efficiency, and robustness.

- The experiments are extensive across synthetic and real-world dynamical systems.

- The paper is well-written and easy to follow.

**Weaknesses:**

- More clarifications on the methodology itself and the experimental setups are appreciated. Please see many questions below.

- One minor issue should be fixed. This paper does not use consistent citation formats. For example, on Page 6, the authors use “NOAA (Reynolds et al., 2002)” and “Williams et al. (2024).”

**Questions:**

- This paper mentioned laptop-level computing. However, I didn’t find the computing configuration in the paper. How does this paper validate this statement? Also, this paper shows the training time in Table 1 without introducing the computing platforms the authors used.

- For the implementation of SINDy loss, do you train the entire measured time sequence as one batch? Suppose you have a sensor measurement s_1 with a time duration of [0,1000]. Do GRU layers take 1000 steps as input without batches? Otherwise, why do you compute the SINDy loss?

- I assume the learned latent dynamics are not sufficiently precise in terms of discovered equations. How do you measure the error propagation of latent discovered physics?

- On Page 4, why does this paper claim that the proposed method is able to avoid spectral bias issues without an encoder? The spectral bias comes from the neural network itself, not just encoders. More discussions are appreciated.

- How do you select the basis functions in the SINDy library? If the latent space is smooth, then you do not need to consider too many basis functions. What are the basis function libraries for the tested cases?

- The authors claim the GRU produces a smoother latent space than LSTM. Do you have an ablation study on that or visualizations?

- I think the superiority of the proposed method comes from the incorporation of the discovered latent physics, as shown in Figure 10. It would be good to have an ablation study to train the same pipeline without SINDy loss to give readers a better sense.

- As shown in Figure 14, how does this paper obtain the original latent space?

---

### Official Review · Reviewer_QHXo · 2024-11-01

**Soundness:** 3
**Presentation:** 3
**Contribution:** 2
**Rating:** 3
**Confidence:** 4

**Summary:**

In the manuscript, a novel algorithm to learn the dynamics of a spatio-temporal process is developed by combining two approaches, SINDy and SHRED. SHRED was proposed to model a spatio-temporal dynamics using a RNN. The authors proposed to use SINDy to regularize the dynamics of the latent state of SHRED. It is shown that the proposed method, SINDy-SHRED, outperforms the baseline methods in some spatio-temporal prediction problem.

**Strengths:**

The proposed method is straightforward. Yet, it seems to make a good prediction of complex physical processes.

**Weaknesses:**

One of the major concerns is that the novelty of the proposed method is limited. It combines two already published methods, SHRED and SINDy. Also, it is not clear if applying SINDy to regularize the dynamics of the latent state is a good choice. The way SINDy is used for the regularization is also not straightforward. See the questions below.

**Questions:**

1. The authors claim that using SINDy the proposed method is "more likely to identify governing physics", and the authors listed some "discovered equations" in the numerical experiments. However, it is unclear how the dynamics in the latent space is related with "governing physics". If a linear sampling function is used as a decoder, as in Kalman filters, it may be possible to related the discovered dynamics to governing physics. But, using a highly flexible nonlinear mapping, correlating the dynamics in the latent state to the governing physics is unlikely.

2. There is an inconsistency in the formulation. In Eqns (1 - 2), the time evolution of $z_t$ is a function of input variable $x_t$, i.e., $\dot{z}(t) = f(x(t))$. Then, in Eqn (3), suddenly, $z_t$ becomes an autonomous system, $\dot{z}(t) = f(z(t))$. Then, for the rest of the manuscript, $z_t$ remains as an autonomous process. This treatment of $z_t$ significantly restricts the class of problems that the proposed method can be applied. The method can be applied only to a stationary process without an exogenous forcing.

3. The ensemble SINDy is a little bit strange. It would have made more sense, if $Z^{GRU}$ is regularized by an ensemble of SINDy functions, e.g, $|| Z^{GRU} - 1/B \sum_i Z^{SINDy}_i ||$. But in the manuscript the authors propose to minimize $\sum_i || Z^{GRU}  -Z^{SINDy}_i ||$. Then, the minimum should be $Z^{SINDy}_1 = \cdots = Z^{SINDy}_B = \hat{Z}^{SINDy} $ that minimizes $|| Z^{GRU}  -\hat{Z}^{SINDy} ||$. In other words, the ensemble SINDy converges to the single SINDy. It is mentioned that one of the purposes of the ensemble SINDy is using "varying levels of sparsity". But it is not explained how $\Xi^i$ is set to "vary the sparsity".

4. One of the purpose of the method is to use a sparse input $X^S$ to reconstruct the whole field $X$. However, to train the model, the whole field data, $X$, is required, which essentially makes a chicken-and-egg problem. The model can reconstruct the whole field, but to do so, the model requires the data for the whole field, which is very difficult to obtain in many real-world problems. Together with the issues with the stationarity of the process that can be modeled (see comment 2), it further limits the problems the problems that can be solved by the proposed method.

5. Algorithm 1 is not very clear to understand. Does line 8 indicates the optimization problem is completely solved for every $i$ and fixed $X_{t-L:t}$?  How is the optimization in Line 8 solved? SGD? Then, it means SGD is run until convergence for every $i$ and every possible $t$? Lines 3 - 7 should be in a subroutine to solve Line 8. What is $n$ in Line 2 by the way?

6. Once trained, SINDy can be used to compute $Z$, instead of GRU? If not, what's the purpose of the sparse discovery? $Z$ does not follow the sparse dynamics anyway.

---

### Official Review · Reviewer_w5AD · 2024-11-03

**Soundness:** 2
**Presentation:** 3
**Contribution:** 2
**Rating:** 3
**Confidence:** 5

**Summary:**

This paper proposed to incorporate SINDy algorithm into the SHRED framework for a more interoperable latent space. The application is mainly in atmospheric science and fluid mechanics. The main contributions are introducing the symbolic understanding of latent space for spatio-temporal dynamics.

**Strengths:**

Originality: The novelty of paper comes from incorporating classic SINDy algorithms into the SHRED framework and demonstrate its effectiveness in several physics dataset.

Quality: The experiment part includes for different cases to investigate the effectiveness of the paper. One case is selected to comprehensively compare with several baseline models.

Clarity: The paper is clear in describing the methodology and the experiment result. It is easy to follow the background and contribution of the proposed work.

Significance: The proposed work could enhance the interoperability of the learnt latent space and improve the prediction accuracy.

**Weaknesses:**

Firstly, the motivation of this paper is not well illustrated. Increasing the latent space interpretability is a important problem. However, why SINDy instead of other symbolic method like symbolic learning is introduced. SINDy's limitations are strong assumptions about the library terms. Usually the library terms are constructed based on some physics intuitions accounting for convection/diffusion. However, the latent space of GRU doesn't actually has physical meaning and it is hard to verify if the derived form could align the underlying physics from the view of interpretability. Some experiments also missing about the convergence of the identified system. For example, if you corrupted the dataset with different noise, will the final analytical form discovered by SINDy keep the same? How was your discovered result change with size of the libraries, e.g., including more terms in the library to show the scalability of the algorithms.

Secondly, the experiments doesn't compare with any symbolic learning, which could justify if SINDy is a better candidate in this problem. Moreover, the comparison is also done for the pendulum case instead of all the cases.

Thirdly, the flow-over cylinder problem has been studied thoroughly by other reduced order methods but there are no relevant comparison. Essentially, the final output of the proposed model is a ROM dynamical model in latent space so it is important to show its performance gain compared to the established methods.

**Questions:**

Questions are just more concrete illustration of the weakness session.

1. Could you justify why SINDy but not other symbolic regression models. Some experiment results would be more convincing in addition to the words

2. Could you show the impact of data noise to the final identified latent space dynamics. To what degree your model could give roughly the same form. Could you also show the impact of the library terms.

3. Could you compare with all your cases with baseline models? My concern is that in Figure 4 and 6 the prediction results don't seem very close to the ground truth data.

4. Could you provide the training cost for all your models?

5. How is your result compared with other ROM model? Could they lead the same/similar form? Is your model more accurate?

6. How sensible your model is respect to the ensemble size?

---

### Note · Authors · 2024-11-14

I have read and agree with the venue's withdrawal policy on behalf of myself and my co-authors.